# HETEROFL: COMPUTATION AND COMMUNICATION EFFICIENT FEDERATED LEARNING FOR HETEROGENEOUS CLIENTS

**Enmao Diao**
Department of Electrical and Computer Engineering
Duke University
Durhm, NC 27705, USA
`enmao.diao@duke.edu`

**Jie Ding**
School of Statistics
University of Minnesota-Twin Cities
Minneapolis, MN 55455, USA
`dingj@umn.edu`

**Vahid Tarokh**
Department of Electrical and Computer Engineering
Duke University
Durhm, NC 27705, USA
`vahid.tarokh@duke.edu`

## ABSTRACT

Federated Learning (FL) is a method of training machine learning models on private data distributed over a large number of possibly heterogeneous clients such as mobile phones and IoT devices. In this work, we propose a new federated learning framework named *HeteroFL* to address heterogeneous clients equipped with very different computation and communication capabilities. Our solution can enable the training of heterogeneous local models with varying computation complexities and still produce a single global inference model. For the first time, our method challenges the underlying assumption of existing work that local models have to share the same architecture as the global model. We demonstrate several strategies to enhance FL training and conduct extensive empirical evaluations, including five computation complexity levels of three model architecture on three datasets. We show that adaptively distributing subnetworks according to clients' capabilities is both computation and communication efficient.

## 1 INTRODUCTION

Mobile devices and the Internet of Things (IoT) devices are becoming the primary computing resource for billions of users worldwide (Lim et al., 2020). These devices generate a significant amount of data that can be used to improve numerous existing applications (Hard et al., 2018). From the privacy and economic point of view, due to these devices' growing computational capabilities, it becomes increasingly attractive to store data and train models locally. Federated learning (FL) (Konečný et al., 2016; McMahan et al., 2017) is a distributed machine learning framework that enables a number of clients to produce a global inference model without sharing local data by aggregating locally trained model parameters. A widely accepted assumption is that local models have to share the same architecture as the global model (Li et al., 2020b) to produce a single global inference model. With this underlying assumption, we have to limit the global model complexity for the most indigent client to train its data. In practice, the computation and communication capabilities of each client may vary significantly and even dynamically. It is crucial to address heterogeneous clients equipped with very different computation and communication capabilities.

In this work, we propose a new federated learning framework called *HeteroFL* to train heterogeneous local models with varying computation complexities and still produce a single global inference model. This **model heterogeneity** differs significantly from the classical distributed machine learning framework where local data are trained with the same model architecture (Li et al., 2020b; Ben-Nun & Hoefler, 2019). It is natural to adaptively distribute subnetworks according to clients'

capabilities. However, to stably aggregate heterogeneous local models to a single global model under various heterogeneous settings is not apparent. Addressing these issues is thus a key component of our work. Our main contributions of this work are three-fold.

- We identify the possibility of model heterogeneity and propose an easy-to-implement framework *HeteroFL* that can train heterogeneous local models and aggregate them stably and effectively into a single global inference model. Our approach outperforms state-of-the-art results without introducing additional computation overhead.

- Our proposed solution addresses various heterogeneous settings where different proportions of clients have distinct capabilities. Our results demonstrate that even when the model heterogeneity changes dynamically, the learning result from our framework is still stable and effective.

- We introduce several strategies for improving FL training and demonstrate that our method is robust against the balanced non-IID statistical heterogeneity. Also, the proposed method can reduce the number of communication rounds needed to obtain state-of-the-art results. Experimental studies have been performed to evaluate the proposed approach.

## 2 RELATED WORK

Federated Learning aims to train massively distributed models at a large scale (Bonawitz et al., 2019). FedAvg proposed by McMahan et al. (2017) is currently the most widely adopted FL baseline, which reduces communication cost by allowing clients to train multiple iterations locally. Major challenges involved in FL include communication efficiency, system heterogeneity, statistical heterogeneity, and privacy (Li et al., 2020b). To reduce communication costs in FL, some studies propose to use data compression techniques such as quantization and sketching (Konečnỳ et al., 2016; Alistarh et al., 2017; Ivkin et al., 2019), and some propose to adopt split learning (Thapa et al., 2020). To tackle system heterogeneity, techniques of asynchronous communication and active sampling of clients have been developed (Bonawitz et al., 2019; Nishio & Yonetani, 2019). Statistical heterogeneity is the major battleground for current FL research. A research trend is to adapt the global model to accommodate personalized local models for non-IID data (Liang et al., 2020), e.g., by integrating FL with other frameworks such as assisted learning (Xian et al., 2020), meta-learning (Jiang et al., 2019; Khodak et al., 2019), multi-task learning (Smith et al., 2017), transfer learning (Wang et al., 2019; Mansour et al., 2020), knowledge distillation (Li & Wang, 2019) and lottery ticket hypothesis (Li et al., 2020a). Nevertheless, these personalization methods often introduce additional computation and communication overhead that may not be necessary. Another major concern of FL is data privacy (Lyu et al., 2020), as model gradient updates can reveal sensitive information (Melis et al., 2019) and even local training data (Zhu et al., 2019; Zhao et al., 2020).

To our best knowledge, what we present is the first work that allows local models to have different architectures from the global model. Heterogeneous local models can allow local clients to adaptively contribute to the training of global models. System heterogeneity and communication efficiency can be well addressed by our approach, where local clients can optimize low computation complexity models and therefore communicate a small number of model parameters. To address statistical heterogeneity, we propose a "Masking Trick" for balanced non-IID data partition in classification problems. We also propose a modification of Batch Normalization (BN) (Ioffe & Szegedy, 2015) as privacy concern of running estimates hinders the usage of advanced deep learning models.

## 3 HETEROGENEOUS FEDERATED LEARNING

### 3.1 HETEROGENEOUS MODELS

Federated Learning aims to train a global inference model from locally distributed data $\{X_1, \ldots, X_m\}$ across $m$ clients. The local models are parameterized by model parameters $\{W_1, \ldots, W_m\}$. The server will receive local model parameters and aggregate them into a global model $W_g$ through model averaging. This process iterates multiple communication rounds and can be formulated as $W_g^t = \frac{1}{m} \sum_{i=1}^{m} W_i^t$ at iteration $t$. At the next iteration, $W_g^t$ is transmitted to a subset of local clients and update their local models as $W_i^{t+1} = W_g^t$.

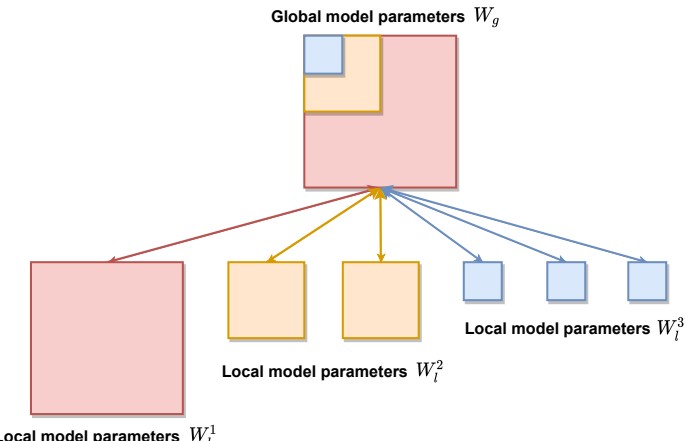

Figure 1: Global model parameters $W_g$ are distributed to $m = 6$ local clients with $p = 3$ computation complexity levels.

In this work, we focus on the relaxation of the assumption that local models need to share the same architecture as the global model. Since our primary motivation is to reduce the computation and communication complexity of local clients, we consider local models to have similar architecture but can shrink their complexity within the same model class. To simplify global aggregation and local update, it is tempting to propose local model parameters to be a subset of global model parameters $W_i^{t+1} \subseteq W_g^t$. However, this raises several new challenges like the optimal way to select subsets of global model parameters, compatibility of the-state-of-art model architecture, and minimum modification from the existing FL framework. We develop Heterogeneous Federated Learning (HeteroFL) to address these issues in the context of deep learning models.

A variety of works show that we can modulate the size of deep neural networks by varying the width and depth of networks (Zagoruyko & Komodakis, 2016; Tan & Le, 2019). Because we aim to reduce the computation complexity of local models, we choose to vary the width of hidden channels. In this way, we can significantly reduce the number of local model parameters, while the local and global model architectures are also within the same model class, which stabilizes global model aggregation.

We demonstrate our method of selecting subsets of global model parameters $W_l$ for a single hidden layer parameterized by $W_g \in \mathbf{R}^{d_g \times k_g}$ in Fig. 1, where $d_g$ and $k_g$ are the output and input channel size of this layer. It is possible to have multiple computation complexity levels $W_l^p \subset W_l^{p-1} \cdots \subset W_l^1$ as illustrated in Fig. 1. Let $r$ be the hidden channel shrinkage ratio such that $d_l^p = r^{p-1} d_g$ and $k_l^p = r^{p-1} k_g$. It follows that the size of local model parameters $|W_l^p| = r^{2(p-1)}|W_g|$ and the model shrinkage ratio $R = \frac{|W_l^p|}{|W_g|} = r^{2(p-1)}$. With this construction, we can adaptively allocate subsets of global model parameters according to the corresponding capabilities of local clients. Suppose that number of clients in each computation complexity level is $\{m_1, \ldots, m_p\}$. Specifically, we perform global aggregation in the following way.

$$W_l^p = \frac{1}{m} \sum_{i=1}^{m} W_i^p, \quad W_l^{p-1} \setminus W_l^p = \frac{1}{m - m_p} \sum_{i=1}^{m - m_p} W_i^{p-1} \setminus W_i^p, \ldots \tag{1}$$

$$W_l^1 \setminus W_l^2 = \frac{1}{m - m_{2:p}} \sum_{i=1}^{m - m_{2:p}} W_i^1 \setminus W_i^2 \tag{2}$$

$$W_g = W_l^1 = W_l^p \cup (W_l^{p-1} \setminus W_l^p) \cup \cdots \cup (W_l^1 \setminus W_l^2) \tag{3}$$

For notational convenience, we have dropped the iteration index $t$. We denote the $W_i^p$ as a matrix/tensor. The $W_g^t[: d_m, : k_m]$ denotes the upper left submatrix with a size of $d_m \times k_m$. Also, $W_g^{p-1,t+1} \setminus W_g^{p,t+1}$ denotes the set of elements included in $W_g^{p-1,t+1}$ but excluded in $W_g^{p,t+1}$.

We exemplify the above equations using Fig. 1. The first part of Equation (1) shows that the smallest part of model parameters (blue, $p = 3$) is aggregated from all the local clients that contain it.

In the second part of Equation (1), the set difference between part $p-1$ (orange) and $p$ (blue) of model parameters is aggregated from local clients with computation complexity level smaller than $p-1$. In Equation (2), the red part of model parameters can be similarly aggregated from $m - m_{2:p} = m_1$ clients. In Equation (3), the global model parameters $W_g^t$ is constructed from the union of all disjoint sets of the partition. In summary, each parameter will be averaged from those clients whose allocated parameter matrix contains that parameter. Thus, a model of an intermediate complexity will have parameters fully averaged with all the other larger models but partially with smaller models (according to the corresponding upper left submatrix).

Several works show that wide neural networks can drop a tremendous number of parameters per layer and still produce acceptable results (Han et al., 2015; Frankle & Carbin, 2018). The intuition is thus to perform global aggregation across all local models, at least on one subnetwork. To stabilize global model aggregation, we also allocate a fixed subnetwork for every computation complexity level. Our proposed inclusive subsets of global model parameters also guarantee that smaller local models will aggregate with more local models.

Thus, small local models can benefit more from global aggregation by performing less global aggregation for part of larger local model parameters. We empirically found that this approach produces better results than uniformly sampled subnetworks for each client or computation complexity level.

### 3.2 STATIC BATCH NORMALIZATION

After global model parameters are distributed to active local clients, we can optimize local model parameters with private data. It is well-known that the latest deep learning models usually adopt Batch Normalization (BN) to facilitate and stabilize optimization. However, classical FedAvg and most recent works avoid BN. A major concern of BN is that it requires running estimates of representations at every hidden layer. Uploading these statistics to the server will cause higher communication costs and privacy issues Andreux et al. (2020) proposes to track running statistics locally.

We highlight an adaptation of BN named as **static Batch Normalization** (sBN) for optimizing privacy constrained heterogeneous models. During the training phase, sBN does not track running estimates and simply normalize batch data. We do not track the local running statistics as the size of local models may also vary dynamically. This method is suitable for HeteroFL as every communication round is independent. After the training process finishes, the server sequentially query local clients and cumulatively update global BN statistics. There exist privacy concerns about calculating global statistics cumulatively and we hope to address those issues in the future work. We also empirically found this trick significantly outperforms other forms of normalization methods including the InstanceNorm (Ulyanov et al., 2016), GroupNorm (Wu & He, 2018), and LayerNorm (Ba et al., 2016) as shown in Table 4 and Table 5.

### 3.3 SCALER

There still exists another cornerstone of our HeteroFL framework. Because we need to optimize local models for multiple epochs, local model parameters at different computation complexity levels will digress to various scales. This known phenomenon was initially discussed by the dropout (Srivastava et al., 2014). To directly use the full model during the inference phase, inverted dropout with dropout rate $q$ scales representations with $\frac{1}{1-q}$ during the training phase. In practice, dropout is usually attached after the activation layer as the selection of subnetworks is performed with masking. Our method directly selects subnetworks from the subsets of global model parameters. Therefore, we append a **Scaler** module right after the parametric layer and before the sBN and activation layers. The Scaler module scales representations by $\frac{1}{r^{p-1}}$ during the training phase. After the global aggregation, the global model can be directly used for inference without scaling. To further illustrate this point, we include a comprehensive ablation study in Tables 4 and 5. A typical linear hidden layer used in our HeteroFL framework can be formulated as

$$y = \phi(\text{sBN}(\text{Scaler}(X_m W_m^p + b_m^p))) \tag{4}$$

where $y$ is the output, $\phi(\cdot)$ denotes a non-linear activation layer, e.g ReLU(), and $W_m^p, b_m^p$ are the weight and bias for local model $m$ at computation complexity level $p$. With all the practical methods mentioned above, we propose the complete pseudo-code for our HeteroFL framework in Algorithm 1. The local capabilities information $L_m$ is an abstraction of the computation and communica-

---

**Algorithm 1:** HeteroFL: Heterogeneous Federated Learning

---

**Input:** Data $X_i$ distributed on $M$ local clients, the fraction $C$ of active clients per
communication round, the number of local epochs $E$, the local minibatch size $B$, the
learning rate $\eta$, the global model parameterized by $W_g$, the channel shrinkage ratio $r$,
and the number of computation complexity levels $P$.

**System executes:**
    Initialize $W_g^0$ and local capabilities information $L_{1:K}$
    **for** each communication round $t = 0, 1, 2, \ldots$ **do**
        $M_t \leftarrow \max(C \cdot M, 1)$
        $S_t \leftarrow$ random set of $M_t$ clients
        **for** each client $m \in S_t$ **in parallel do**
            Determine computation complexity level $p$ based on $L_m$
            $r_m \leftarrow r^{(p-1)}, d_m \leftarrow r_m d_g, k_m \leftarrow r_m k_g$
            $W_m^t \leftarrow W_g^t[: d_m, : k_m]$
            $W_m^{t+1} \leftarrow \text{ClientUpdate}(m, r_m, W_m^t)$
        **end**
        **for** each computation complexity level $p$ **do**
            $W_g^{p-1,t+1} \setminus W_g^{p,t+1} \leftarrow \frac{1}{M_t - M_{p:P,t}} \sum_{i=1}^{M_t - M_{p:P,t}} W_i^{p-1,t+1} \setminus W_i^{p,t+1}$
        **end**
        $W_g^{t+1} \leftarrow \bigcup_{p=1}^{P} W_g^{p-1,t+1} \setminus W_g^{p,t+1}$
        Update $L_{1:K}, \eta$ (Optional)
    **end**
    Query representation statistics from local clients (Optional)
**ClientUpdate** $(m, r_m, W_m)$**:**
    $B_m \leftarrow$ Split local data $X_m$ into batches of size $B$
    **for** each local epoch $e$ from 1 to $E$ **do**
        **for** batch $b_m \in B_m$ **do**
            $W_m \leftarrow W_m - \eta \nabla \ell(W_m, r_m; b_m)$
        **end**
    **end**
    Return $W_m$ to server

---

tion capabilities of a local client $m$. Once this information is communicated to the server, the server can know the model complexity that should be allocated to the client. We can also optionally update learning rates to facilitate optimization and local capabilities information if changing dynamically.

## 4 EXPERIMENTAL RESULTS

We trained over 600 individual models for exploring and demonstrating the effectiveness of our method. We experimented with MNIST and CIFAR10 image classification tasks and the WikiText2 language modeling task (LeCun et al., 1998; Krizhevsky et al., 2009; Merity et al., 2016; Devlin et al., 2018).

Our experiments are performed with three different models including a CNN for MNIST, a preactivated ResNet (PreResNet18) (He et al., 2016) for CIFAR10 and a Transformer (Vaswani et al., 2017) for WikiText2. We replace BN in CNN and PreResNet18 with our proposed sBN, and attach the Scaler module after each convolution layer. To study federated optimization, we adopt data partition the same as in (McMahan et al., 2017; Liang et al., 2020). We have $100$ clients, and the fraction $C$ of active clients per communication round is $0.1$ throughout our experiments. For IID data partition, we uniformly assign the same number of data examples for each client. For balanced non-IID data partition, we assume that the label distribution is skewed, where clients will only have examples at most from two classes and the number of examples per class is balanced. We note that there exist other kinds of non-IID data partition, e.g., the unbalanced non-IID data partition where clients may hold unbalanced labeled dataset and the feature distribution skew where clients may hold different features. We conduct a masked language modeling task with a 15% masking rate and uniformly

assign balanced data examples for each client. It needs to point out that each client will roughly have 3000 different words in their local dataset, while the total vocabulary size is 33278. The details regarding hyperparameters and model architecture can be found in Table 6 of the Appendix.

To study the effectiveness of our proposed HeteroFL framework, we construct five different computation complexity levels $\{a, b, c, d, e\}$ with the hidden channel shrinkage ratio $r = 0.5$. We have tried various shrinkage ratios, and we found that it is most illustrative to use the discrete complexity levels 0.5, 0.25, 0.125, and 0.0625 (relative to the most complex model). For example, model 'a' has all the model parameters, while models 'b' to 'e' have the effective shrinkage ratios 0.5, 0.25, 0.125, and 0.0625. We note that the complexity of 'e' is close to a logistic regression model. Our experiments indicated that the ratio can be arbitrary between $(0, 1]$ and dynamically change. In practice, using a dictionary of discrete complexity levels are convenient for coordination purposes.

Each local client is assigned an initial computation complexity level. We annotate *Fix* for experiments with a fixed assignment of computation complexity levels, and *Dynamic* for local clients uniformly sampling computation complexity levels at each communication round. We perform distinct experiments for *Fix* and *Dynamic* assignments. All the figures are based on the *Fix* scenario, where we considered models of different sizes, and each client is allocated with a fixed size. All the tables are based on the *Dynamic* scenario, where we randomly vary the allocation of clients' model complexity, and the ratio of the number of weak learners is fixed to 50%.

The x-axis of figures represents the average model parameters. When 10% clients use the model 'a' and 90% use the model 'e', the average number of model parameters is $0.1 \times$ (size of model 'a') $+ 0.9 \times$ (size of model 'e'). We interpolate this partition from 10% to 100% with step size 10% to demonstrate the effect of proportionality of clients with various computation complexity levels.

To demonstrate the effect of dynamically varying computation and communication capabilities, we uniformly sample from various combinations of computation complexity levels. For example, model 'a-b-c-d-e' means that we uniformly sample from all possible available levels for every active client at each communication round. We show the number of model parameters, FLOPs, and Space (MB) to indicate the computation and communication requirements of our methods. For example, since we uniformly sample levels, model $a - e$ calculates computational metrics by averaging those of model $a$ and $e$. The ratio is calculated between the number of parameters of a given model with respect to its 100% global model.

We compare our results to other baseline methods like Standalone, FedAvg, and LG-FedAvg gathered from (Liang et al., 2020). Standalone means there is no communication between clients and the server. In our experimental studies, we considered more complex models compared with the existing work. In particular, the baseline models in LG-FedAvg used MLP (on MNIST) and CNN (on CIFAR10). In terms of the number of parameters, our models 'a-e' (on MNIST) and 'b-e' (on CIFAR10) are comparable with those baselines. In terms of the FLOPs, our model 'd-e' (on MNIST and CIFAR10) can be compared with those baselines. The single-letter models 'a', 'b', 'c', 'd', 'e' are our implementations of the FedAvg equipped with the sBN and Masking CrossEntropy. The takeaway of Table 2 is that a weak learner that can only train a small model 'e' (on CIFAR10)(77.09%) can boost its performance to 'c-e' (86.88%), 'b-e' (89.10%), or 'a-e' (90.29%), which are close to the scenario where all the learners are strong, namely c(87.55%), b(89.82%), or a(91.99%). In particular, in 'c-e', 'b-e', or 'a-e', half of the clients are trained with larger models 'c', 'b', or 'a', while the other half are trained with the model 'e'. Only the aggregated global models 'c', 'b', or 'a' are used during the testing stage. Although weak clients train smaller models 'e', they will test with the largest models 'c', 'b', or 'a' to gain better performance.

Full results including other possible combinations can be found in appendix in Table 7-9. Finally, our method is robust to dynamically varying model complexities. It is worth noting that our method does not incur any additional computation overhead and can be readily adapted to existing applications.

We also perform experiments for balanced non-IID data partition and provide a simple trick to achieve comparable results. As mentioned earlier, most state-of-the-art results ofbalanced non-IID datasets suggest the personalization of local models to achieve better local results (Smith et al., 2017; Liang et al., 2020; Li et al., 2020a). Here, the *Local* results assume that the training data distribution and test data distribution for each local client are the same, and assign zero probability for those classes that are not presented to a client during training. The *Global* results were calculated from the global model applied to the test data directly. The *Local* results were cumulatively averaged from

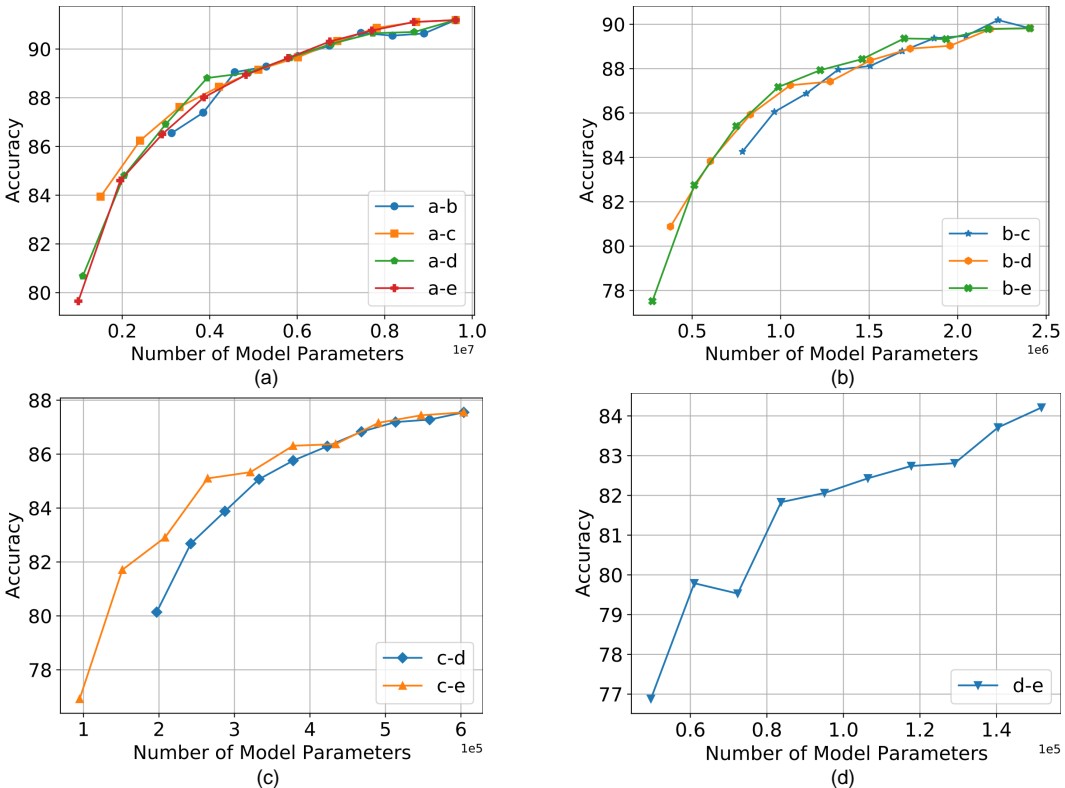

Figure 2: Interpolation experimental results for CIFAR10 (IID) dataset between global model complexity ((a) a, (b) b, (c) c, (d) d) and various smaller model complexities.

the performance of each data example on each local client. Zhao et al. (2018) showed that the failure of non-IID FL is related to the weight divergence among local model parameters trained locally for many iterations. The weight divergence mostly occurs in the last classification layer of networks.

Thus, instead of a full Cross-Entropy Loss for all classes, we are motivated to train each local model only with their corresponding classes. In this way, each local model will train a sub-task given locally available label information. Specifically, we mask out the output of the model before passing it Cross-Entropy Loss, which we named as **Masked Cross-Entropy Loss**. We experimented with several different ways of masking, we find replacing the last layer outputs that are not associated with local labels with zero achieves both stable and comparable local and global results. When aggregating local model parameters, we do not aggregate the untrained parameters in the last classification layers. Either the server can infer this information implicitly, or the local clients can report which classes they have to the server explicitly. We provide a comprehensive ablation study in Tables 5. The results show that Masked Cross-Entropy Loss significantly improve local performance and moderately global performance of balanced non-IID data partition task. Since our primary focus is to address model heterogeneity, we leave the analysis of this trick to future work. We show the results of *Fix* experiments in appendix in Fig. 5-8. *Dynamic* non-IID results are also included in Table 1-3. The results show that our method performs comparably to those with personalized local models. Our method is readily adaptable, free of computation overhead, and only rely on the single global model for testing local and global results. It allows local clients to switch to another subtask simply by changing its mask without querying the server for others' personalized models.

We show the learning curves of 50% *Fix* and *Dynamic* assignments in appendix in Fig. 9-11. The learning curves show that the optimization of HeteroFL for the IID dataset is stable and efficient. Our method achieves better results with a fewer number of communication rounds, e.g., 800 for Heterofl and 1800 for LG-FedAvg (Liang et al., 2020). We empirically discover gradient clipping stabilizes the optimization of HeteroFL as it prevents small models from gradient explosion. We can therefore adopt a universal learning rate for heterogeneous local models. It is also perceivable

| Model | Ratio | Parameters | FLOPs | Space (MB) | Accuracy | | |
|---|---|---|---|---|---|---|---|
| | | | | | IID | Non-IID | |
| | | | | | | Local | Global |
| a | 1.00 | 1.6 M | 80.5 M | 5.94 | 99.53 | 99.85 | 98.92 |
| a-e | 0.50 | 782 K | 40.5 M | 2.98 | 99.46 | 99.89 | 98.96 |
| a-b-c-d-e | 0.27 | 416 K | 21.6 M | 1.59 | 99.46 | 99.85 | 98.29 |
| b | 1.00 | 391 K | 20.5 M | 1.49 | 99.53 | 99.87 | 99.10 |
| b-e | 0.51 | 199 K | 10.4 M | 0.76 | 99.51 | 99.67 | 98.51 |
| b-c-d-e | 0.33 | 131 K | 6.9 M | 0.50 | 99.52 | 99.88 | 98.99 |
| c | 1.00 | 99 K | 5.3 M | 0.38 | 99.35 | 99.56 | 96.34 |
| c-e | 0.53 | 53 K | 2.9 M | 0.20 | 99.39 | 99.79 | 97.27 |
| c-d-e | 0.44 | 44 K | 2.4 M | 0.17 | 99.31 | 99.76 | 97.85 |
| d | 1.00 | 25 K | 1.4 M | 0.10 | 99.17 | 99.86 | 97.86 |
| d-e | 0.63 | 16 K | 909 K | 0.06 | 99.19 | 99.63 | 97.70 |
| e | 1.00 | 7 K | 400 K | 0.03 | 98.66 | 99.07 | 92.84 |
| Standalone (Liang et al., 2020) | 1.00 | 633 K | 1.3 M | 2.42 | 86.24 | 98.72 | 30.41 |
| FedAvg (Liang et al., 2020) | 1.00 | 633 K | 1.3 M | 2.42 | 97.93 | 98.20 | 98.20 |
| LG-FedAvg (Liang et al., 2020) | 1.00 | 633 K | 1.3 M | 2.42 | 97.93 | 98.54 | 98.17 |

Table 1: Results of combination of various computation complexity levels for MNIST dataset. Full results can be found in Table 7.

| Model | Ratio | Parameters | FLOPs | Space (MB) | Accuracy | | |
|---|---|---|---|---|---|---|---|
| | | | | | IID | Non-IID | |
| | | | | | | Local | Global |
| a | 1.00 | 9.6 M | 330.2 M | 36.71 | 91.19 | 92.38 | 56.88 |
| a-e | 0.50 | 4.8 M | 165.9 M | 18.43 | 90.29 | 92.10 | 59.11 |
| a-b-c-d-e | 0.27 | 2.6 M | 88.4 M | 9.78 | 88.83 | 92.49 | 61.64 |
| b | 1.00 | 2.4 M | 83.3 M | 9.19 | 89.82 | 93.83 | 55.45 |
| b-e | 0.51 | 1.2 M | 42.4 M | 4.67 | 89.10 | 90.68 | 59.81 |
| b-c-d-e | 0.33 | 801 K | 27.9 M | 3.05 | 87.92 | 91.90 | 59.10 |
| c | 1.00 | 604 K | 21.2 M | 2.30 | 87.55 | 91.09 | 55.12 |
| c-e | 0.53 | 321 K | 11.3 M | 1.22 | 86.88 | 91.83 | 63.47 |
| c-d-e | 0.44 | 265 K | 9.4 M | 1.01 | 85.79 | 91.49 | 55.42 |
| d | 1.00 | 152 K | 5.5 M | 0.58 | 84.21 | 90.77 | 61.13 |
| d-e | 0.63 | 95 K | 3.5 M | 0.36 | 82.93 | 90.89 | 56.16 |
| e | 1.00 | 38 K | 1.5 M | 0.15 | 77.09 | 89.62 | 54.16 |
| Standalone (Liang et al., 2020) | 1.00 | 1.8 M | 3.6 M | 6.88 | 16.90 | 87.93 | 10.03 |
| FedAvg (Liang et al., 2020) | 1.00 | 1.8 M | 3.6 M | 6.88 | 67.74 | 58.99 | 58.99 |
| LG-FedAvg (Liang et al., 2020) | 1.00 | 1.8 M | 3.6 M | 6.88 | 69.76 | 91.77 | 60.79 |

Table 2: Results of combination of various computation complexity levels for CIFAR10 dataset. Full results can be found in Table 8.

that aggregation of model parameters trained with non-IID data makes the optimization less stable. Results of *Dynamic* show that global aggregation of dynamically varying computation complexities is stable.

## 5 CONCLUSIONS AND FUTURE WORK

We propose Heterogeneous Federated Learning (HeteroFL), which shows the possibility of coordinatively training local models much smaller than a global model to produce a single global inference model. Our experiments show that FL can be made more practical by introducing HeteroFL

| Model | Ratio | Parameters | FLOPs | Space (MB) | Perplexity |
|-------|-------|-----------|-------|-----------|-----------|
| a | 1.00 | 19.3 M | 1.4 B | 73.49 | 3.37 |
| a-e | 0.53 | 10.2 M | 718.6 M | 38.86 | 3.75 |
| a-b-c-d-e | 0.37 | 7.2 M | 496.6 M | 27.55 | 3.55 |
| b | 1.00 | 9.1 M | 614.8 M | 34.74 | 3.46 |
| b-e | 0.56 | 5.1 M | 342.0 M | 19.49 | 3.90 |
| b-c-d-e | 0.46 | 4.2 M | 278.7 M | 16.07 | 3.64 |
| c | 1.00 | 4.4 M | 290.1 M | 16.92 | 3.62 |
| c-e | 0.62 | 2.8 M | 179.7 M | 10.57 | 3.89 |
| c-d-e | 0.58 | 2.6 M | 166.7 M | 9.85 | 3.66 |
| d | 1.00 | 2.2 M | 140.7 M | 8.39 | 3.83 |
| d-e | 0.75 | 1.7 M | 105.0 M | 6.31 | 3.90 |
| e | 1.00 | 1.1 M | 69.3 M | 4.23 | 7.41 |

Table 3: Results of combination of various computation complexity levels for WikiText2 dataset. Full results can be found in Table 9.

| Model | Normalization | Scaler | Accuracy IID | |
|-------|--------------|--------|-------|---------|
| | | | MNIST | CIFAR10 |
| a | None | N/A | 99.2 | 81.3 |
| | IN | | 99.5 | 87.7 |
| | GN | | 99.5 | 81.0 |
| | LN | | 99.5 | 77.3 |
| | sBN | | **99.6** | **91.7** |
| e | None | N/A | 98.6 | 58.1 |
| | IN | | 97.4 | 66.4 |
| | GN | | 98.7 | 62.6 |
| | LN | | 98.6 | 53.7 |
| | sBN | | **98.7** | **77.0** |
| a-e | None | ✗ | 99.5 | 80.1 |
| | sBN | | 99.0 | 90.1 |
| | None | ✓ | 99.2 | 80.4 |
| | IN | | 99.5 | 86.6 |
| | GN | | 99.5 | 76.0 |
| | LN | | 99.4 | 71.7 |
| | sBN | | **99.5** | **90.1** |

Table 4: Ablation Study of IID scenarios. The single-letter models 'a' and 'e' are FedAvg equiped with various normalization methods. The sBN significantly outperforms other existing normalization methods, including the InstanceNorm (IN), GroupNorm (GN) (the number of group G=4), and LayerNorm (LN). Scaler is used for HeteroFL to train models of different sizes and moderately improve the results.

and sBN and Mased Cross-Entropy Loss, as HeteroFL fully exploits local clients' capabilities and achieves better results with a fewer number of communication rounds. We demonstrate our results with various model architectures, including CNN, PreResNet18, and Transformer, and show that our method is robust to statistical heterogeneity and dynamically varying local capabilities. A future direction is to distinct model classes as well as model heterogeneity. Also, the proposed methods may be emulated to address heterogeneous few-shot learning, multi-modal learning, and multi-task learning.

ACKNOWLEDGMENTS

This work was supported by the Office of Naval Research (ONR) under grant number N00014-18-1-2244, and the Army Research Office (ARO) under grant number W911NF-20-1-0222.

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

## A  APPENDIX

The appendix contains supplementary experimental results. In Table 5 we show the ablation study of Non-IID experiments. Compared to results of IID experiments shown in Table 4, Table .5 shows the ablation study Maksed CrossEntropy which is shown beneficial for balanced Non-IID data partition. In Table 6, we show the hyperparameters adopted in our experiments. In Figure 3 and 4, we show the *Fix* complexity assignments of MNIST and WikiText with iid data partition experiments. From Figure 6 to 8, we show the *Fix* complexity assignments for all balanced Non-IID data partition experiments. Figure 9 to 11, we show the learning curve of experiments with *Dyanmic* complexity assignments. The complete results for all experiments with *Dyanmic* complexity assignments can be found in Table 7, 8, and 9.

| Model | Normalization | Scaler | Masked CrossEntropy | Accuracy Non-IID | | | |
| --- | --- | --- | --- | --- | --- | --- | --- |
| | | | | MNIST | | CIFAR10 | |
| | | | | Local | Global | Local | Global |
| | None | N/A | ✗ | 97.4 | 97.4 | 42.6 | 42.8 |
| | sBN | | | 99.4 | **99.4** | 53.4 | 53.7 |
| a | None | | | 99.7 | 95.6 | 91.7 | 58.5 |
| | IN | | | 99.8 | 98.7 | 88.4 | 43.7 |
| | GN | N/A | ✓ | 99.7 | 98.3 | 91.2 | 58.2 |
| | LN | | | 99.8 | 98.3 | 89.9 | 54.2 |
| | sBN | | | **99.9** | 98.6 | **92.1** | **59.2** |
| | None | N/A | ✗ | 96.2 | 96.0 | 38.9 | 38.2 |
| | sBN | | | 90.1 | 90.1 | 40.7 | 40.4 |
| e | None | | | **99.5** | **96.5** | 86.6 | 48.9 |
| | IN | | | 98.5 | 89.8 | 83.7 | 37.0 |
| | GN | N/A | ✓ | 99.2 | 92.2 | 83.5 | 36.7 |
| | LN | | | 99.3 | 94.0 | 82.6 | 40.0 |
| | sBN | | | 99.3 | 94.2 | **90.1** | **52.9** |
| | None | ✗ | ✗ | 96.8 | 96.9 | 37.8 | 37.4 |
| | sBN | | | 99.2 | 99.2 | 41.0 | 41.3 |
| | None | ✗ | ✓ | 99.4 | 95.7 | 89.1 | 52.8 |
| | sBN | | | 99.8 | 98.0 | 90.7 | 57.7 |
| a-e | None | ✓ | ✗ | 97.3 | 97.3 | 34.6 | 34.4 |
| | sBN | | | 99.3 | **99.3** | 46.0 | 46.7 |
| | None | | | 99.5 | 95.8 | 90.3 | 55.6 |
| | IN | | | 99.8 | 98.7 | 87.0 | 34.4 |
| | GN | ✓ | ✓ | 99.5 | 96.2 | 88.7 | 49.7 |
| | LN | | | 99.5 | 96.2 | 78.5 | 25.0 |
| | sBN | | | **99.8** | 98.2 | **92.8** | **60.4** |

Table 5: Ablation Study of Non-IID scenarios. The Masked CrossEntropy is used for Non-IID experiments. It significantly improves the local performance and moderately improves global performance. Single letter model 'a' and 'e' are FedAvg equipped with various normalization methods. The sBN significantly outperforms other existing normalization methods, including InstanceNorm (IN), GroupNorm (GN) (the number of group G=4), and LayerNorm (LN). Scaler is used for HeteroFL to train models of different sizes and moderately improve the results.

| Data | | MNIST | CIFAR10 | WikiText2 |
|---|---|---|---|---|
| Model | | CNN | PreResNet18 | Transformer |
| Hidden size | | [64, 128, 256, 512] | [64, 128, 256, 512] | [512, 512, 512, 512] |
| Local Epoch $E$ | | 5 | 5 | 1 |
| Local Batch Size $B$ | | 10 | 10 | 100 |
| Optimizer | | | SGD | |
| Momentum | | | 0.9 | |
| Weight decay | | | 5.00E-04 | |
| Learning rate $\eta$ | | 0.01 | 0.1 | 0.1 |
| Communication rounds | IID | 200 | 400 | 100 |
| | non-IID | 400 | 800 | 200 |
| Decay schedule (0.1) | IID | [100] | [150, 250] | [25, 50] |
| | non-IID | [200] | [300, 500] | [50, 100] |
| Embedding Size | | | | 256 |
| Number of heads | | | | 8 |
| Dropout | | N/A | | 0.2 |
| Sequence length | | | | 64 |

Table 6: Hyperparameters and model architecture used in our experiments.

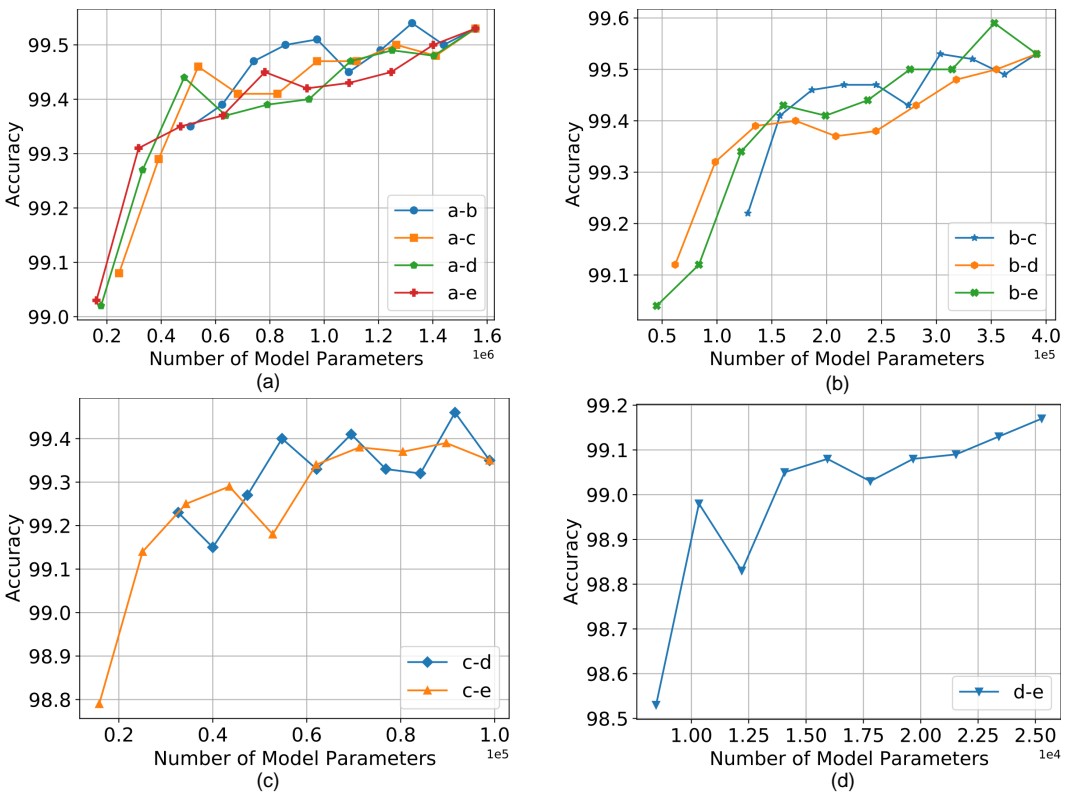

Figure 3: Interpolation experimental results for MNIST (IID) dataset between global model complexity ((a) a, (b) b, (c) c, (d) d) and various smaller model complexities.

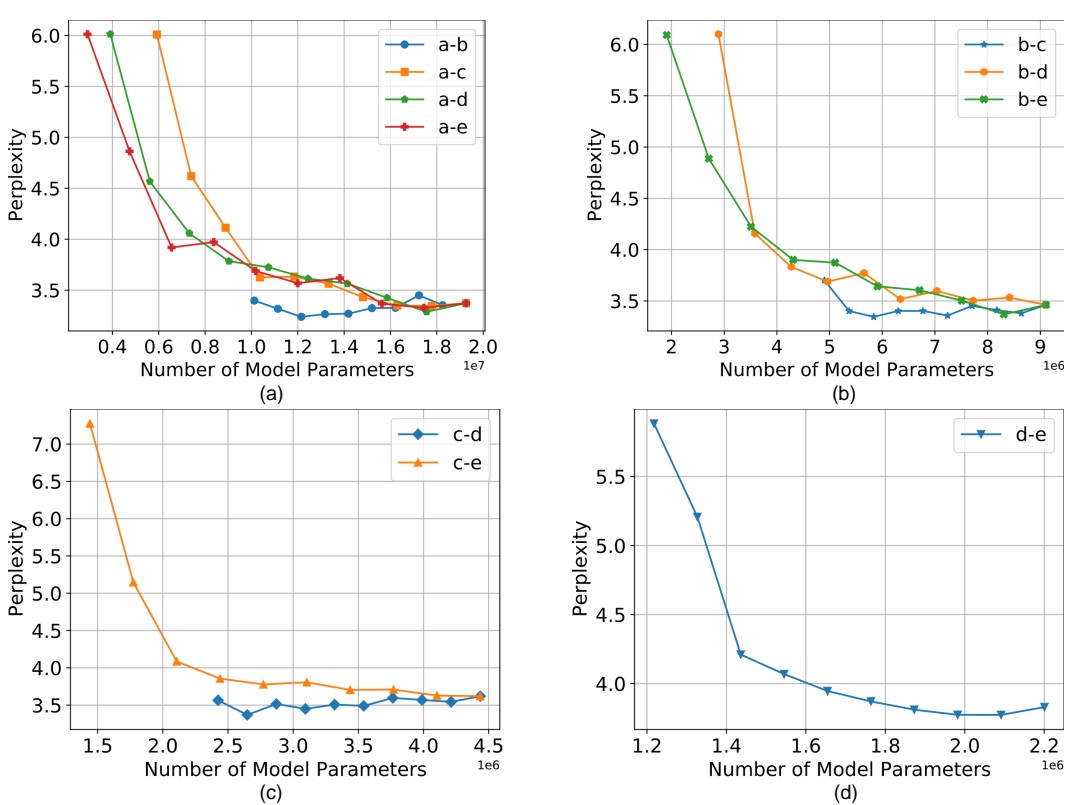

Figure 4: Interpolation experimental results for WikiText2 (IID) dataset between global model complexity ((a) a, (b) b, (c) c, (d) d) and various smaller model complexities.

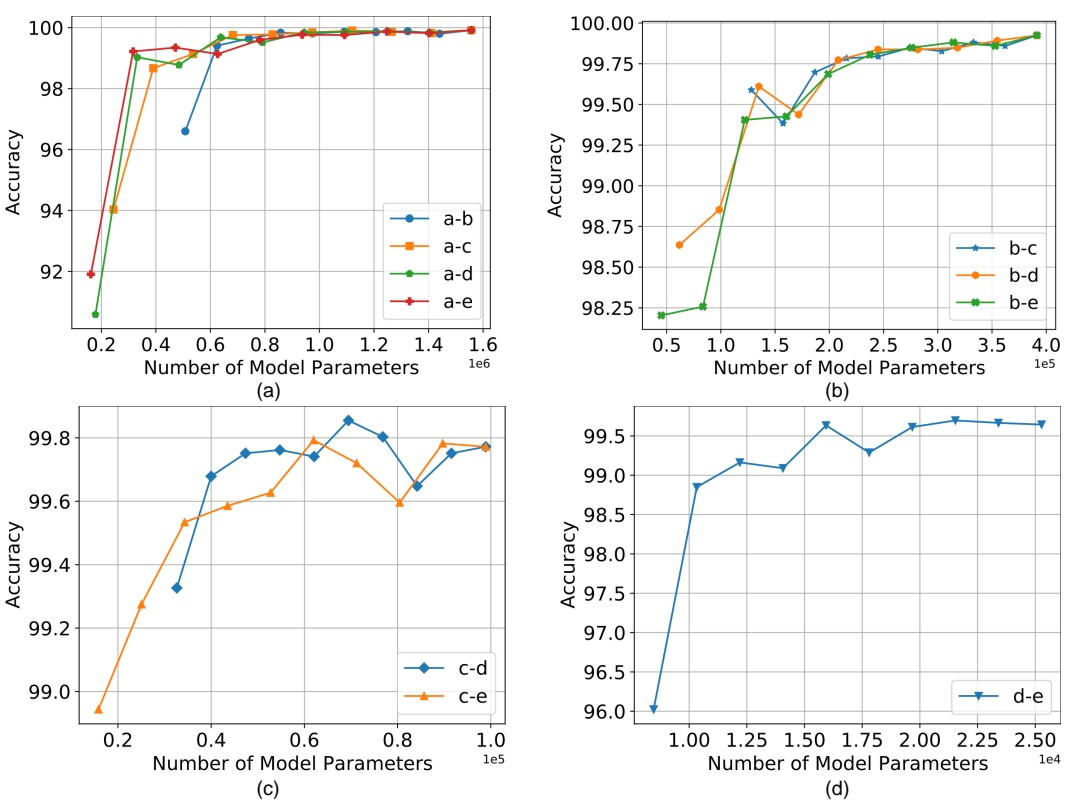

Figure 5: Interpolation experimental results for MNIST (non-IID, Local) dataset between global model complexity ((a) a, (b) b, (c) c, (d) d) and various smaller model complexities.

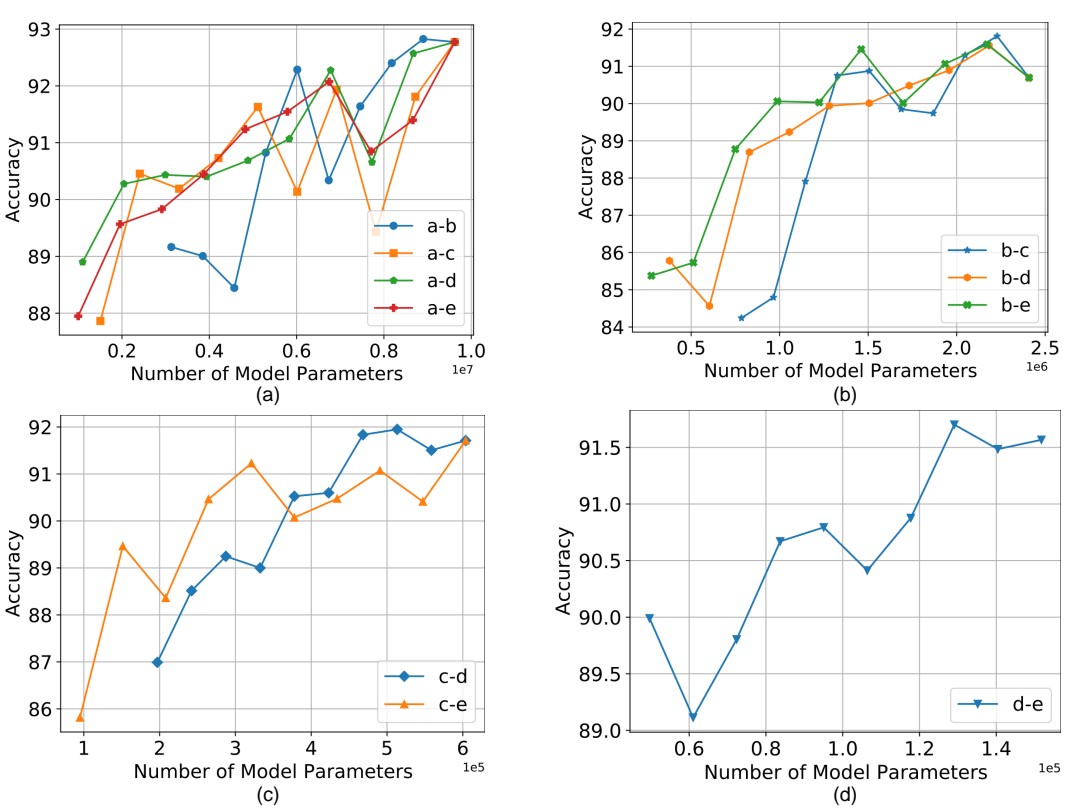

Figure 6: Interpolation experimental results for CIFAR10 (non-IID, Local) dataset between global model complexity ((a) a, (b) b, (c) c, (d) d) and various smaller model complexities.

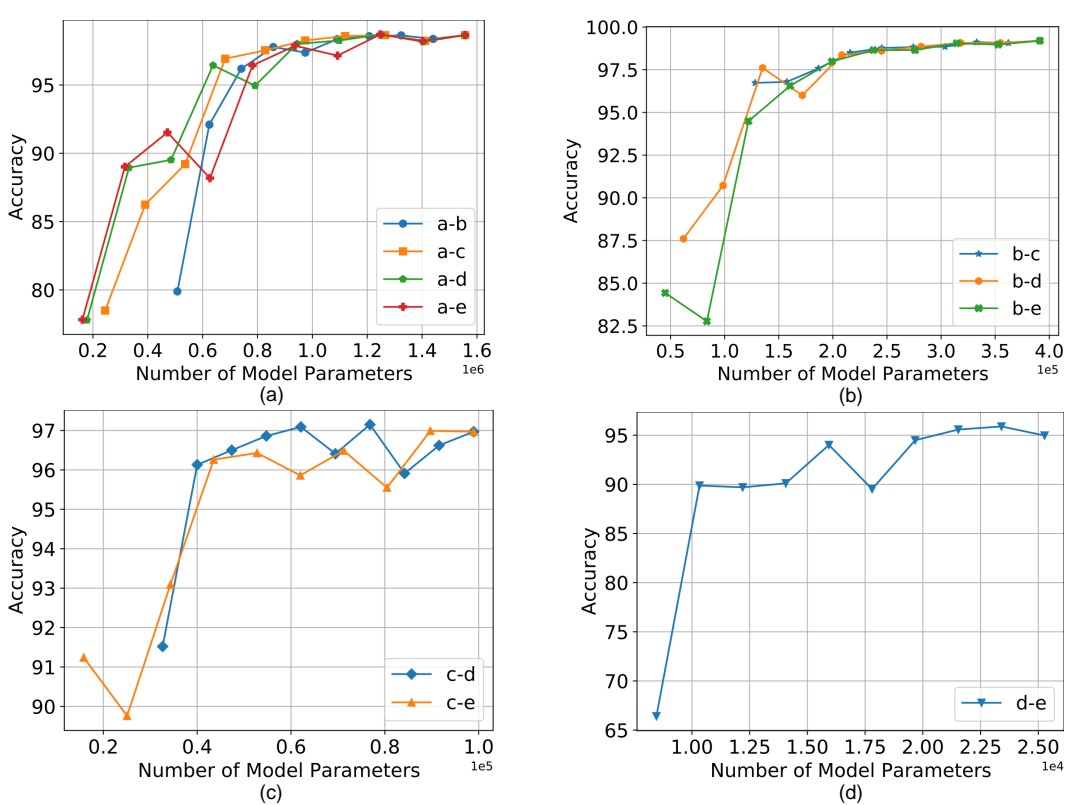

Figure 7: Interpolation experimental results for MNIST (non-IID, Global) dataset between global model complexity ((a) a, (b) b, (c) c, (d) d) and various smaller model complexities.

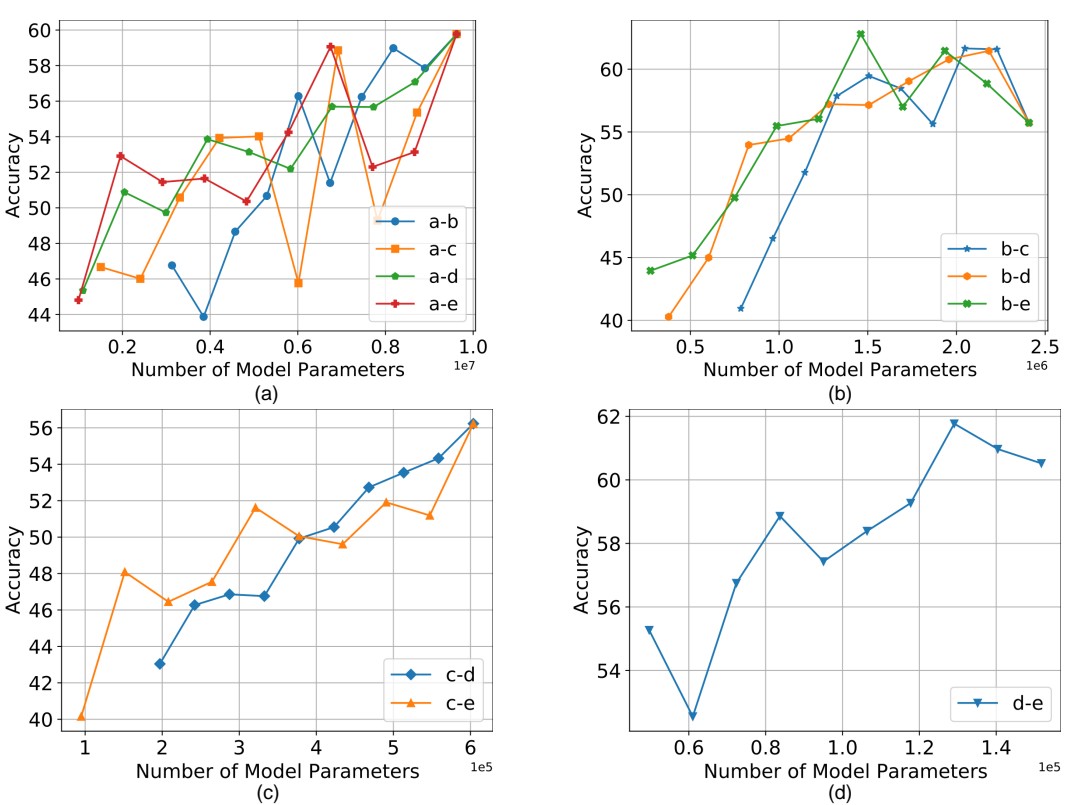

Figure 8: Interpolation experimental results for CIFAR10 (non-IID, Global) dataset between global model complexity ((a) a, (b) b, (c) c, (d) d) and various smaller model complexities.

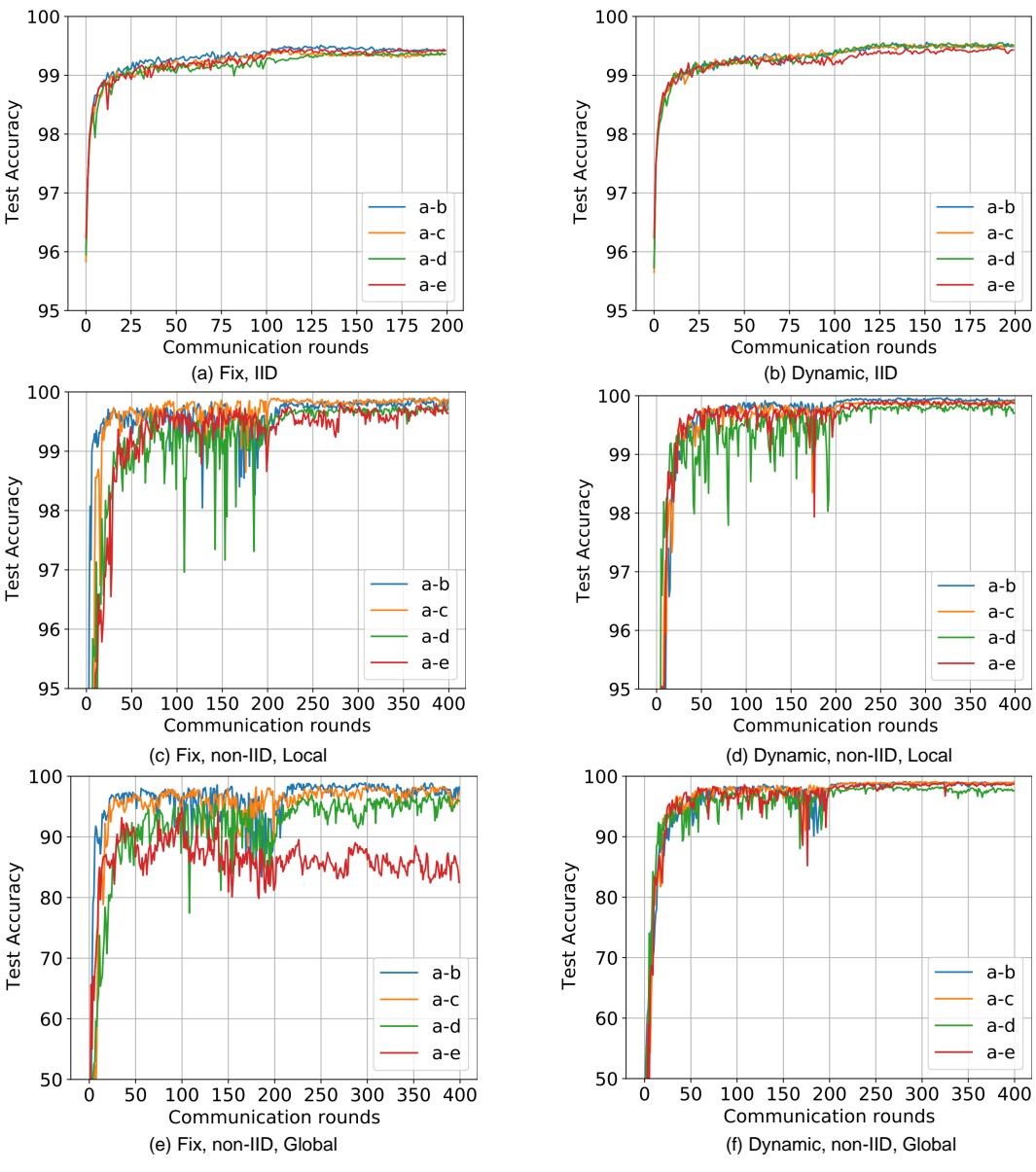

Figure 9: Learning curves of MNIST datasets with 50% Fix and Dynamic computation complexity assignments.

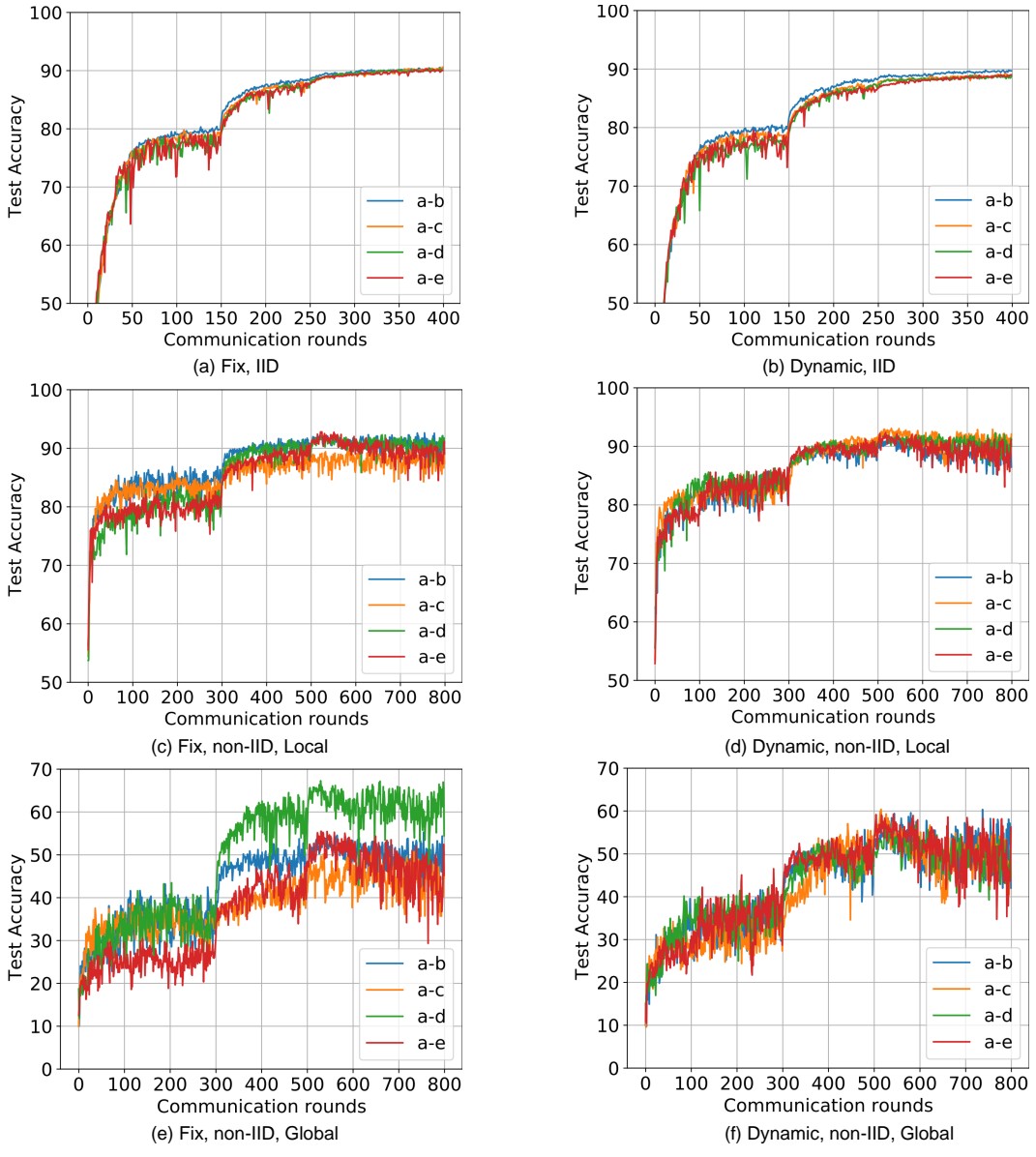

Figure 10: Learning curves of CIFAR10 datasets with 50% Fix and Dynamic computation complexity assignments.

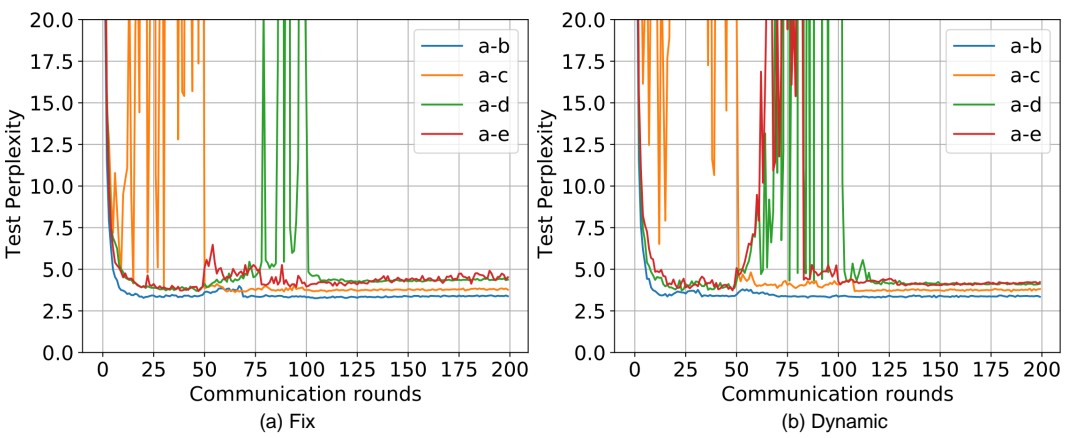

Figure 11: Learning curves of WikiText2 datasets with 50% Fix and Dynamic computation complexity assignments.

| Model | Ratio | Parameters | FLOPs | Space (MB) | Accuracy | | |
|---|---|---|---|---|---|---|---|
| | | | | | IID | Non-IID | |
| | | | | | | Local | Global |
| a | 1.00 | 1.6 M | 80.5 M | 5.94 | 99.53 | 99.85 | 98.92 |
| a-b | 0.63 | 974.1 K | 50.5 M | 3.72 | 99.54 | 99.96 | 99.10 |
| a-c | 0.53 | 827.9 K | 42.9 M | 3.16 | 99.52 | 99.89 | 99.12 |
| a-d | 0.51 | 791.1 K | 41.0 M | 3.02 | 99.54 | 99.81 | 98.37 |
| a-e | 0.50 | 781.7 K | 40.5 M | 2.98 | 99.46 | 99.89 | 98.96 |
| a-b-c | 0.44 | 682.4 K | 35.4 M | 2.60 | 99.53 | 99.90 | 98.72 |
| a-b-d | 0.42 | 657.8 K | 34.1 M | 2.51 | 99.52 | 99.78 | 98.02 |
| a-b-e | 0.42 | 651.6 K | 33.8 M | 2.49 | 99.54 | 99.95 | 98.92 |
| a-c-d | 0.36 | 560.4 K | 29.1 M | 2.14 | 99.57 | 99.95 | 99.34 |
| a-c-e | 0.36 | 554.1 K | 28.7 M | 2.11 | 99.52 | 99.94 | 98.43 |
| a-d-e | 0.34 | 529.6 K | 27.4 M | 2.02 | 99.57 | 99.80 | 98.92 |
| a-b-c-d | 0.33 | 518.1 K | 26.9 M | 1.98 | 99.54 | 99.80 | 99.03 |
| a-b-c-e | 0.33 | 513.4 K | 26.7 M | 1.96 | 99.46 | 99.69 | 97.53 |
| a-b-d-e | 0.32 | 495.0 K | 25.7 M | 1.89 | 99.49 | 99.85 | 98.66 |
| a-c-d-e | 0.27 | 421.9 K | 21.9 M | 1.61 | 99.54 | 99.84 | 98.80 |
| a-b-c-d-e | 0.27 | 415.8 K | 21.6 M | 1.59 | 99.46 | 99.85 | 98.29 |
| b | 1.00 | 391.4 K | 20.5 M | 1.49 | 99.53 | 99.87 | 99.10 |
| b-c | 0.63 | 245.1 K | 12.9 M | 0.94 | 99.49 | 99.87 | 99.05 |
| b-d | 0.53 | 208.3 K | 11.0 M | 0.79 | 99.44 | 99.85 | 98.95 |
| b-e | 0.51 | 199.0 K | 10.4 M | 0.76 | 99.51 | 99.67 | 98.51 |
| b-c-d | 0.44 | 171.9 K | 9.1 M | 0.66 | 99.54 | 99.84 | 98.98 |
| b-c-e | 0.42 | 165.6 K | 8.7 M | 0.63 | 99.51 | 99.85 | 98.20 |
| b-d-e | 0.36 | 141.1 K | 7.4 M | 0.54 | 99.48 | 99.89 | 98.72 |
| b-c-d-e | 0.33 | 130.5 K | 6.9 M | 0.50 | 99.52 | 99.88 | 98.99 |
| c | 1.00 | 98.9 K | 5.3 M | 0.38 | 99.35 | 99.56 | 96.34 |
| c-d | 0.63 | 62.1 K | 3.4 M | 0.24 | 99.38 | 99.92 | 99.05 |
| c-e | 0.53 | 52.8 K | 2.9 M | 0.20 | 99.39 | 99.79 | 97.27 |
| c-d-e | 0.44 | 43.6 K | 2.4 M | 0.17 | 99.31 | 99.76 | 97.85 |
| d | 1.00 | 25.3 K | 1.4 M | 0.10 | 99.17 | 99.86 | 97.86 |
| d-e | 0.63 | 15.9 K | 909.5 K | 0.06 | 99.19 | 99.63 | 97.70 |
| e | 1.00 | 6.6 K | 400.5 K | 0.03 | 98.66 | 99.07 | 92.84 |
| Standalone (Liang et al., 2020) | 1.00 | 633.2 K | 1.3 M | 2.42 | 86.24 | 98.72 | 30.41 |
| FedAvg (Liang et al., 2020) | 1.00 | 633.2 K | 1.3 M | 2.42 | 97.93 | 98.20 | 98.20 |
| LG-FedAvg (Liang et al., 2020) | 1.00 | 633.2 K | 1.3 M | 2.42 | 97.93 | 98.54 | 98.17 |

Table 7: Results of combination of various computation complexity levels for MNIST dataset.

| Model | Ratio | Parameters | FLOPs | Space (MB) | Accuracy | | |
|---|---|---|---|---|---|---|---|
| | | | | | IID | Non-IID | |
| | | | | | | Local | Global |
| a | 1.00 | 9.6 M | 330.2 M | 36.71 | 91.19 | 92.38 | 56.88 |
| a-b | 0.63 | 6.0 M | 206.8 M | 22.95 | 90.60 | 91.35 | 59.93 |
| a-c | 0.53 | 5.1 M | 175.7 M | 19.50 | 90.59 | 92.83 | 60.25 |
| a-d | 0.51 | 4.9 M | 167.9 M | 18.64 | 90.28 | 91.78 | 56.54 |
| a-e | 0.50 | 4.8 M | 165.9 M | 18.43 | 90.29 | 92.10 | 59.11 |
| a-b-c | 0.44 | 4.2 M | 144.9 M | 16.07 | 89.70 | 90.41 | 54.16 |
| a-b-d | 0.42 | 4.1 M | 139.7 M | 15.49 | 89.98 | 90.29 | 51.79 |
| a-b-e | 0.42 | 4.0 M | 138.4 M | 15.35 | 89.79 | 90.79 | 62.17 |
| a-c-d | 0.36 | 3.5 M | 119.0 M | 13.20 | 89.47 | 89.82 | 53.13 |
| a-c-e | 0.36 | 3.4 M | 117.6 M | 13.05 | 89.35 | 93.59 | 57.30 |
| a-d-e | 0.34 | 3.3 M | 112.4 M | 12.48 | 88.75 | 91.11 | 56.74 |
| a-b-c-d | 0.33 | 3.2 M | 110.1 M | 12.19 | 89.33 | 91.32 | 54.50 |
| a-b-c-e | 0.33 | 3.2 M | 109.1 M | 12.09 | 89.37 | 92.52 | 61.56 |
| a-b-d-e | 0.32 | 3.1 M | 105.1 M | 11.65 | 89.40 | 91.80 | 56.78 |
| a-c-d-e | 0.27 | 2.6 M | 89.6 M | 9.93 | 88.42 | 91.50 | 62.15 |
| a-b-c-d-e | 0.27 | 2.6 M | 88.4 M | 9.78 | 88.83 | 92.49 | 61.64 |
| b | 1.00 | 2.4 M | 83.3 M | 9.19 | 89.82 | 93.83 | 55.45 |
| b-c | 0.63 | 1.5 M | 52.3 M | 5.75 | 89.00 | 89.96 | 52.29 |
| b-d | 0.53 | 1.3 M | 44.4 M | 4.88 | 89.18 | 91.78 | 51.07 |
| b-e | 0.51 | 1.2 M | 42.4 M | 4.67 | 89.10 | 90.68 | 59.81 |
| b-c-d | 0.44 | 1.1 M | 36.7 M | 4.02 | 88.35 | 92.79 | 58.09 |
| b-c-e | 0.42 | 1.0 M | 35.3 M | 3.88 | 87.98 | 91.98 | 58.28 |
| b-d-e | 0.36 | 866.3 K | 30.1 M | 3.30 | 88.06 | 91.94 | 54.02 |
| b-c-d-e | 0.33 | 800.7 K | 27.9 M | 3.05 | 87.92 | 91.90 | 59.10 |
| c | 1.00 | 603.8 K | 21.2 M | 2.30 | 87.55 | 91.09 | 55.12 |
| c-d | 0.63 | 377.8 K | 13.4 M | 1.44 | 86.75 | 91.58 | 54.61 |
| c-e | 0.53 | 321.1 K | 11.3 M | 1.22 | 86.88 | 91.83 | 63.47 |
| c-d-e | 0.44 | 264.6 K | 9.4 M | 1.01 | 85.79 | 91.49 | 55.42 |
| d | 1.00 | 151.8 K | 5.5 M | 0.58 | 84.21 | 90.77 | 61.13 |
| d-e | 0.63 | 95.1 K | 3.5 M | 0.36 | 82.93 | 90.89 | 56.16 |
| e | 1.00 | 38.4 K | 1.5 M | 0.15 | 77.09 | 89.62 | 54.16 |
| Standalone (Liang et al., 2020) | 1.00 | 1.8 M | 3.6 M | 6.88 | 16.90 | 87.93 | 10.03 |
| FedAvg (Liang et al., 2020) | 1.00 | 1.8 M | 3.6 M | 6.88 | 67.74 | 58.99 | 58.99 |
| LG-FedAvg (Liang et al., 2020) | 1.00 | 1.8 M | 3.6 M | 6.88 | 69.76 | 91.77 | 60.79 |

Table 8: Results of combination of various computation complexity levels for CIFAR10 dataset.

| Model | Ratio | Parameters | FLOPs | Space (MB) | Perplexity |
|---|---|---|---|---|---|
| a | 1.00 | 19.3 M | 1.4 B | 73.49 | 3.37 |
| a-b | 0.74 | 14.2 M | 991.4 M | 54.12 | 3.31 |
| a-c | 0.62 | 11.8 M | 829.0 M | 45.20 | 3.71 |
| a-b-c | 0.57 | 10.9 M | 757.6 M | 41.72 | 3.42 |
| a-d | 0.56 | 10.7 M | 754.3 M | 40.94 | 3.74 |
| a-b-d | 0.53 | 10.2 M | 707.8 M | 38.87 | 3.53 |
| a-e | 0.53 | 10.2 M | 718.6 M | 38.86 | 3.75 |
| a-b-e | 0.51 | 9.8 M | 684.0 M | 37.49 | 3.47 |
| b | 1.00 | 9.1 M | 614.8 M | 34.74 | 3.46 |
| a-b-c-d | 0.45 | 8.8 M | 603.4 M | 33.39 | 3.61 |
| a-c-d | 0.45 | 8.6 M | 599.6 M | 32.93 | 4.08 |
| a-b-c-e | 0.44 | 8.5 M | 585.5 M | 32.34 | 3.50 |
| a-c-e | 0.43 | 8.3 M | 575.8 M | 31.54 | 3.65 |
| a-b-d-e | 0.41 | 7.9 M | 548.2 M | 30.21 | 3.64 |
| a-d-e | 0.39 | 7.5 M | 526.0 M | 28.70 | 4.02 |
| a-b-c-d-e | 0.37 | 7.2 M | 496.6 M | 27.55 | 3.55 |
| b-c | 0.74 | 6.8 M | 452.4 M | 25.83 | 3.45 |
| a-c-d-e | 0.35 | 6.8 M | 467.0 M | 25.76 | 3.92 |
| b-d | 0.62 | 5.7 M | 377.7 M | 21.57 | 3.70 |
| b-c-d | 0.58 | 5.2 M | 348.5 M | 20.02 | 3.47 |
| b-e | 0.56 | 5.1 M | 342.0 M | 19.49 | 3.90 |
| b-c-e | 0.54 | 4.9 M | 324.7 M | 18.63 | 3.46 |
| c | 1.00 | 4.4 M | 290.1 M | 16.92 | 3.62 |
| b-c-d-e | 0.46 | 4.2 M | 278.7 M | 16.07 | 3.64 |
| b-d-e | 0.45 | 4.1 M | 274.9 M | 15.79 | 3.92 |
| c-d | 0.75 | 3.3 M | 215.4 M | 12.66 | 3.46 |
| c-e | 0.62 | 2.8 M | 179.7 M | 10.57 | 3.89 |
| c-d-e | 0.58 | 2.6 M | 166.7 M | 9.85 | 3.66 |
| d | 1.00 | 2.2 M | 140.7 M | 8.39 | 3.83 |
| d-e | 0.75 | 1.7 M | 105.0 M | 6.31 | 3.90 |
| e | 1.00 | 1.1 M | 69.3 M | 4.23 | 7.41 |

Table 9: Results of combination of various computation complexity levels for WikiText2 dataset.

