# OpenReview forum: "HeteroFL: Computation and Communication Efficient Federated Learning for Heterogeneous Clients"
_ICLR.cc/2021/Conference — ICLR 2021 Poster_

### Official Review · AnonReviewer4 · 2020-10-21
**Good idea, but bad execution and a lot of confusion**

**Rating:** 7
**Confidence:** 4

**Review:**

UPDATE:
The authors have consistently improved their argumentation over the course of the review and addressed my concerns sufficiently. I have increased my score and recommend acceptance.


ORIGINAL REVIEW:
The authors propose three elements:
A way to approach model heterogeneity across clients with different resource constraints, a 'masking trick' for approaching non-iid-ness and a modification to BatchNormalization in the Federated Learning setting.

In order to receive differently sized models that still allow to be aggregated systematically on a central server, the authors propose to deterministically prune away neurons/feature maps of NNs to effectively create networks of different widths. While a more powerful end-device receives, computes with and updates all feature maps, a less powerful device computes only with a fraction of those feature maps. This idea seems novel and interesting to me and I commend the extensive experimental study.

For the proposed 'static BN', the authors propose to not worry about running estimates across all clients until convergence of the model. I disagree with both, the proposition that 'static BN' is something new compared to normal BatchNormalization, nor that it inherently solves the issue with BN in the federated setup. Firstly, BN in section 3.1 of the original paper suggests using moving averages only as a means to estimate test-time performance during training. At convergence, the final model requires re-computing statistics on the whole dataset, identical to what is proposed in this work. Secondly, apart from the problem of finding a global statistics estimate at the end of training, the usage of mini-batch statistics during training is used as an alternative to the global estimates since the global estimates are too expensive to compute during training. In centralised training, a random mini-batch represents the global data-set well enough. In non-iid data settings, a random mini-batch from a client does not serve as a good estimate of the global data statistics, only for the statistics of its local data-set.
I therefore do not understand how the proposed solution of 'sBN does not track running estimates and simply normalize [sic] batch data' addresses the issue that a client-level mini-batch does not represent global statistics.
A common approach to dealing with BN is to simply replace BN with, for example Group Normalization (https://arxiv.org/pdf/1910.00189.pdf) (Also see Section 5 for a discussion of the issues of BN in FL).

I am therefore asking the authors to explain the exact differences of sBN to normal BN with respect to re-estimation of global statistics at the end of training and to elaborate on the issue of using mini-batch statistics as an estimate of global data-set statistics. If indeed there is a difference to normal BN, I would like to see explicit experimental results that compare normal BN to sBN and, ideally, to GroupNormalization as a way to circumvent the BN issue.
Maybe I am misunderstanding in the sense that the 'Masking Trick' also somehow alleviates the non-iid data issues with BN. If that is the case, I would like to see an explicit ablation study that distinguishes between the two.


With respect to non-iid data and the proposed 'masking trick' the authors cite Zhao et al. stating that the weight divergence mostly occurs in the last classification layer of networks. Inspecting Figure 2 of that paper, this conclusion can be drawn only for one of the three experiments at display. I agree that this is a minor point and the proposed trick sounds reasonable and interesting to me. In oder to see its effectiveness, however, there needs to be an ablation study with and without that trick. I cannot find such an experiment in the paper. Furthermore it should be stated that label skew is just one of the possible sources of non-iid-ness in FL. Lastly, the authors mention that the masking trick 'allows local clients to switch to another subtask simply by changing its mask...'. I have troubles understanding what is implied here. From a client's perspective there is only its local label-distribution. If a client is assumed to be new to the federation of clients, the new client would receive the un-masked global model presumably. In which setting would a client require a new mask (from another client?)

Experimental Evaluation:
The authors present a large range of experiments for different scenarios and levels of heterogeneity between clients.
I do have issues understanding the results precisely though. The authors do not mention what the x-axes in Figure 2 represent. Is the y-axis local or global accuracy? In combination with Tables 1 and 2, I am confused. If Standalone and FedAvg have 633K parameters respectively, how does a 100% model a (first row in Table 1) have 1.6M parameters? Presumably, experiments were conducted with the same full, 100% CNN model architecture. Alternatively, the hyper parameters in Table 4 in the Appendix do not make sense to me. An alternative interpretation would be that these are the amount of parameters communicated until convergence - but then again the hyperparamters are inconclusive and additionally, the space requirements of 100% model a should still be the same as FedAvg. Or does this column describe the amount of communication at 32bit float precision?
If the authors chose a different architecture for their baselines, then the results are inconclusive.
I am assuming that 'Standalone' refers to no communication between clients, but that needs to be specified!
Also I am assuming that 'Local' assigns zero probability on $p(y=c|x)$ for those classes $c$ that are not present on a client during training. Again, this is not explicitly specified. Are the reported results averaged across clients? Are they weighted by the amount of data in the local test-sets?  In the conclusion, the authors state that their method achieves better results with fewer number of communication rounds. I can no-where see a comparison of communication rounds.

The authors mention two scenarios: Fix and Dynamic and explicitly say 'We annotate Fix for experiments with a fixed assignment of computation complexity levels and Dynamic for local clients uniformly sampling ...". I cannot find this annotation anywhere and am therefore confused which setting the results correspond to.

I am missing one axis of evaluation: In a heterogeneous (Fixed) setting with, for example, 50% a and 50% level e, how is the average local performance on devices with model a and model e separately. In the text, the authors describe 100% model e achieves 77.09% accuracy and a 50-50 mix with model a achieves 89%. But that does tell me nothing about how much the (weak) clients with model e improved through the increased power in these other 50% devices. In the next sentence the authors claim that 'HeteroFL can boost client's performance with low computation and communication capabilities'. But reporting the average could also allow for the conclusion that only clients with higher compute power and a larger model achieve higher performance.

Conclusion
The paper proposed three elements, a heterogeneous modelling approach, sBN and the masking trick. Apart from confusion in motivation and explanation, the experimental section requires most attention in my opinion. Things are simply very unclear to me.  Terms, axes and results need to be properly discussed. Furthermore, the effects of HeteroFL, sBN and the masking-trick need to be independently studied, otherwise no conclusion can be drawn on the effectiveness of the individual ideas. I would recommend the authors to re-focus their paper on the heterogeneous training idea alone and leave sBN, which I don't understand at the moment, and the non-iid remedy trough masking to another paper.

I believe that the idea of dynamically adjusting the model width to the local compute capabilities in the way the authors present it is promising. However I need to be convinced that less powerful clients can meaningfully contribute to the global model and have higher performance compared to training a small-sized global model in the first place. I encourage the authors to revisit their motivation for elements of this work (HeteroFL, sBN and masking), refocus, and fix their experimental discussion. I believe that there is enough merit to this idea and paper to be accepted with major effort during the rebuttal.

---

> ### Author Response · Authors · 2020-11-17
> **Response**
>
> Thank you for the constructive comments. We address each of the concerns below.
> 1. To address your concern, we have provided a comprehensive ablation study in the revision, including the comparison with InstanceNorm, GroupNorm, and LayerNorm (Tables 4 and 5 in the revised paper).
> The main difference between sBN and BN is that we do not track running estimates and just use mini-batch statistics to ensure the feature before meeting the affine parameters has zero mean and unit variance. Although the BN work suggests computing the overall statistics after the training is completed, for practical implementations of BN, we use the tracked running estimates during the inference stage. Transmitting running statistics as used in the vanilla BN will likely to cause privacy issues. That is why we advocate the one-time transmission of feature statistics of a finalized model.
> We believe the principal merit of sBN is not the running estimates approximating the global statistics. The sBN facilitates and stablizes the training because it standardizes the feature before meeting the affine parameters. Our result shows that running statistics or even global statistics do not seem necessary during training. Even the data are non-iid split, mini-batch normalization will ensure the features to have zero mean and unit variance. Thus, the local training and global aggregation of model parameters are stable even for models of different sizes. When the global statistics are calculated during the testing, the features meeting affine parameters are still approximately zero mean and unit variance.
> 2. To address your first concern in point 2, we have provided another ablation study in the revision, including the comparison between masking and no-masking (Table 5 in the revised paper).
> Second, we have now mentioned in the revision that label skew is just one of the possible sources of non-iid-ness in FL. We mentioned that the masking trick allows local clients to easily switch to another subtask by changing their mask. We elaborate more on it.
> After training, the clients will receive a global aggregated model. Depending on each local client's label distribution, the client can choose to mask which part of their output. It is also possible for local clients to switch to another label distribution. In the personalized FL method, the local clients will have to query the server for a new personalized model for the new label distribution. In our case, the local client can mask the output based on the new label distribution and still produce a good performance because we only have one single global inference model.

---

> > ### Author Response · Authors · 2020-11-17
> > **Response continue**
> >
> > 3. In the IID case, we do not distinguish between local and global results because we only have one aggregated model tested against the whole test dataset. The X-axis of figures represents the average model parameters. When 10\% clients use the model 'a' and 90\% use the model 'e', the average number of model parameters is 0.1*(size of model 'a') + 0.9*(size of model 'e'). We interpolate this partition from 10\% to 100\% with step size 10\%.
> > The tabulated results of Standalone, FedAvg, and LG-FedAvg were state-of-the-art results gathered from the LG-FedAvg paper. In our experimental studies, we considered more complex models compared with the existing work. The baseline models in LG-FedAvg used MLP (on MNIST) and CNN (on CIFAR10). In terms of the number of parameters, our models 'a-e' (on MNIST) and 'b-e' (on CIFAR10) are comparable with those baselines. In terms of the FLOPs, our model 'd-e' (on MNIST and CIFAR10) can be compared with those baselines. The single-letter models 'a', 'b', 'c', 'd', 'e' are our implementations of the FedAvg equipped with the sBN and Masking CrossEntropy. For all the above model configurations, our latest ablation study shows that our the sBN and Masking CrossEntropy can significantly improve the results.
> > The takeaway of Table 2 is that a weak learner that can only train a small model 'e' (on CIFAR10)(77\%) can boost its performance to 'c-e' (86.88\%), 'b-e' (89.10\%), or 'a-e' (90.29\%), which are close to the scenario where all the learners are strong, namely c(87.55\%),  b(89.82\%), or a(91.99\%).
> > In particular, in 'c-e', 'b-e', or 'a-e', half of the clients are trained with larger models 'c', 'b', or 'a', while the other half are trained with the model 'e'. During the testing stage, only the aggregated global models 'c', 'b', or 'a' are used. Although weak clients train smaller models 'e', they will test with the largest models 'c', 'b', or 'a' to better perform.
> > We have now clarified the definition of Standalone and Local in the revision. As to your interpretations stated in ''I am assuming that 'Standalone' refers to no communication between clients, but that needs to be specified! Also, I am assuming that 'Local' assigns zero probability for those classes that are not present on a client during training,'' yes, that is what we meant. We have clarified them in the revision.
> > For the questions, ''Are the reported results averaged across clients? Are they weighted by the amount of data in the local test sets?''
> > The reported results were not averaged across clients but averaged across each example. Our reported results were categorized into two settings, namely the global and the local. The global results were calculated from the global model applied to the test data directly. The local results were cumulatively averaged from the performance of each data example on each local client.
> > We have mentioned the comparison of communication rounds by texts in the current revision. In particular, the learning curves shown in Figures 10 of the appendix indicate that we only need to train 800 communication rounds for CIFAR10, while the LG-FedAvg needs about 1800 rounds.
> > 4.  All the figures are based on the Fix scenario, where we considered models of different sizes, and each client is allocated with a fixed size.
> > All the tables are based on the Dynamic scenario, where we randomly vary the allocation of clients' model complexity, and the ratio of the number of weak learners is fixed to 50\%.
> >  A 50-50 mix with model 'a' can achieve 90.29\%,  indicating that we can boost the weak clients. Recall that the inference model is the aggregated model 'a' but not the small subnetwork 'e' inside the model 'a'. For example, the 100\% model 'e' produces a single model 'e' and achieves 77.09\%. The 100\% model 'a' produces a single model 'a' and achieves 91.19\%. The 50-50 model 'a-e' produces a single model 'a' and achieves 90.29\%.
> > Thus, the weak clients can only train a subnetwork 'e' but benefit from the performance of the aggregated model 'a'.
> > 5.  In summary, the benefits are for weak clients with small local models. In the classical FedAvg, if some weak clients can only train a small model 'e' due to computation and communication constraints, all the other clients have to train the same small model 'e' to average the model parameters from weak clients. The weak clients were the bottleneck of the overall performance of Federate learning. In our proposed solution, each local client can train models corresponding to their personalized computation and communication capabilities. Then, the aggregated global model is deployed for each local client for future inference.

---

> > > ### Comment · AnonReviewer4 · 2020-11-23
> > > **Thx for the rebuttal**
> > >
> > > This rebuttal has cleared my confusions. I know understand how to interpret the experimental results. Crucially, I was under the assumption that a 'weak' client does inherently not have the power to execute a 'larger' model. Therefore the 'dynamic' setting, as well as the evaluation did not make much sense to me. I understand now, that you assume a 'weak' client to still be able to execute the global model at test time.
> > >
> > > I still believe it would be very interesting to see the performance of a weak client executing a weak model at test time. Does a weak client benefit from executing only a small part of a bigger model?
> > >
> > > I see the ablation studies that I have requested, thank you.
> > >
> > > In light of these updates, I raise my score to 5.
> > >
> > > In order to raise my score to 6, I need a better argumentation for sBN. I don't feel like the authors have addressed my concerns here.
> > >
> > > My understanding of sBN is: Compute BN statistics locally once at the end of training and communicate those to the server, where they are averaged.
> > > I do not agree with the novelty claim here. BN never *required* computing and maintaining running averages in the first place. BN in federated learning never required transmitting these running averages to the server at any point. Running averages were always used as a proxy to track model performance during training.
> > > Arguably, running averages are necessary in research only to plot learning curves on hold-out-data and to enable early stopping. Even then, there is no principled reason to not just re-compute statistics on the whole training set for each of these occasions, where it not for the huge computational overhead. Removing this overhead is the only reason for using running averages.
> > > Even for different model sizes during training, there is no reason to maintain running averages, so no problem is 'solved' by proposing to not do it.
> > >
> > > What existing BN *does* claim, is recomputing statistics once at the end of training, instead of using running averages, since it is a well-worth trade-off in computational overhead to do. Crucially, sBN claims to do exactly that, so no novelty here.
> > >
> > > Furthermore, I don't disagree that private data might be reconstructable with running averages, however, private information can also leak through the final communication of statistics. Having access to multiple such statistics across different layers plus having access to the model architecture that generated them plus the potentially small size of the local data-set still poses a (as of yet unquantified) privacy risk in federated learning. With (s)BN, differential privacy guarantees might be not applicable any more.
> > >
> > > Finally, why BN in FL is not the normalisation method of choice is because the average of local statistics at the server is a poor approximation of the global statistics. This can be directly seen in Table 5 of your experiments. I have to admit that I am very surprised about the performance of GN, LN, IN in the global setting. In my own experiments, I see different results, but of course, details matter.
> > >
> > > Finally, I the masking trick. I need to understand based on what logic a client will request which 'other' mask. In which situation would a client a) need a new mask, b) know that it needs a new mask, c) know which mask it needs? Also, would the masking trick alleviate label skew that is less binary it its class assignment? There are many works in this field which create non-iid-ness based on drawing from a dirichlet distribution. I can provide references if required. What would happen if each client has at least a single data-point per class? Label skew would still be present significantly, but the masking trick would not work.
> > > While I see the power of the masking trick to the presented experimental scenarios, I don't see it generalising to label skew in general.
> > > If the authors make this limitation clear in their submission, and fix their claims/discussion surrounding sBN, I will raise my score to 7. Should they only discuss the masking trick without fixing sBN, I will remain at 6.
> > >
> > >
> > >
> > >
> > > Section 3 seems to be empty. Something went wrong with the section enumeration

---

> > > > ### Author Response · Authors · 2020-11-23
> > > > **Response**
> > > >
> > > > Thank you for the constructive comments. We address each of the concerns below.
> > > > 1.  We consider a 'dynamic' setting because some local clients may have varying computation capabilities. Our experiments aim to demonstrate that it is possible for HeteroFL to adapt dynamically-varying computation capabilities. For example, the clients may use a computer or mobile device for other purposes like web browsing, gaming, and video streaming while also participating in the training. In this scenario, local computation capabilities can vary significantly during the training phase.
> > > > A weak client executing a weak model at test time is indeed an interesting direction. We may add the results of testing with small models in the next revision or future work.
> > > >
> > > > 2.  We agree that the novelty of sBN is limited. In the latest revision, we highlight that sBN is an adaptation of BN because we have to train models of different sizes in HeteroFL. Please see our response (3) below for the calculation of final statistics.
> > > > We agree that the final communication of statistics leaks some private information. However, we believe that the leaked private information is not enough for reconstructing the original dataset. For example, there exist many trained models online, and we usually use them for transfer learning. To our best knowledge, we are not aware of a method that can reconstruct the original dataset simply by knowing one snapshot of model parameters (including BN statistics) and architecture. Nevertheless, the privacy leakage in our context has not been quantified, and we believe it is an interesting future work.
> > > >
> > > > 3.  Thank you for pointing out the difficulty of averaging local statistics. Indeed, pooling local statistics and averaging them is a poor approximation of the global statistics. We have to clarify that instead of pooling local statistics, we sequentially and cumulatively calculate the global statistics. In particular, after the training process is finished, the server sends the model parameters to the first local client. The first local client sends its BN statistics to the server. Then, the server sends the model parameters, the BN statistics, and the sample size from the first client to the second local client so that the second local client can cumulatively update the BN statistics. Then, the second local client does the same thing as the first client, and so on.
> > > >
> > > > 4.  We illustrate the usage of the switch of masking with an example. Suppose that there are two clients A and B. Client A learns to classify 10 classes of dogs, and client B learns to classify 10 classes of cats. They may jointly learn a network to classify 20 classes of dogs and cats. When client A knows that the testing data belongs to either dog or cat, it can switch the mask to classify corresponding sub-classes.
> > > > The masking trick will alleviate label skew for more than two classes. We experimented with balanced two-class non-iid data (where each client has two class labels) because it is the most skewed case. In general, the partition into two-class non-iid data produces worse results than the partition into more than two classes. The two-class scenario was also studied in the original FedAvg and LG-FedAvg papers.
> > > > We have pointed out the limitation of the masking trick in the revision. In particular, our method assumes the non-iid distribution is balanced, meaning that each class has a similar amount of data. The masking trick does not aim to solve the unbalanced non-iid data partition problem in general.
> > > >
> > > > 5.  We have fixed the section enumeration issue.

---

> > > > > ### Comment · AnonReviewer4 · 2020-11-24
> > > > > **Almost there**
> > > > >
> > > > > About 2:
> > > > > Still this point remains in the text: 'Thus, this method greatly reduces the risk of leaking private data', which is somewhat of strange argument given that the original BN never requires the sending of running averages (and thus would leak private data). I agree with the general sentiment of not sending running averages, but I do not agree that this is a novel insight. I concede that it is always easy in hindsight to say: "Well, obviously you don't send running averages", when no-one has formulated it explicitly before, but still, I find myself thinking that I would never have considered it, also before reading this paper.
> > > > > I also don't know of a method that would reconstruct the 'original dataset' (not an expert here), however there are many more 'attacks' that one could perform, such as, for example, membership inference (which is often the target of differential privacy in FL). Now, to which degree your method enables any kind of attacks with respect to alternative method is of course not clear and generally speaking hard to quantify (prove).
> > > > > Case in point, however, your point 3 offers additional interesting insight into your method, especially wrt. privacy. Generally speaking, the number of data-points per client is revealed only to the server, not other clients. The current discussion of sBN is good enough for me to raise my score, although, as you can see, I'm not really happy with it. Even if I believe the methodology to be sound!
> > > > >
> > > > > I remain in my assessment that point 4 is somewhat contrived - how does client A know that it is presented with an out-of-distribution example (a cat), when it has never seen one? The server, or the human user, would need to intervene and provide that information along with the new mask. Alternatively, use no mask and make use of the global model in the first place.
> > > > >
> > > > > Furthermore, you haven't addressed (in the paper), my point about general label-skew. The sentence remains:
> > > > > 'To address statistical heterogeneity, we propose a “Masking Trick” and demonstrate that personalization is unnecessary for non-IID data in classification problems. ' This is wrong on two accounts: There is more non-IID type of data than label skew and the additional caveat of uniform distribution of remaining classes. I believe your rebuttal comment here to be a bit strange, since we are (hopefully) not optimising for very specific benchmarks (the two-class problem) and bake that prior information into our algorithm. It is fine if we assume label skew as the type of non-iid-ness, but then the additional assumption of having uniformity of remaining classes seems very, very narrow to me.
> > > > >
> > > > > As such, I remain in my assessment that your method does not contribute much to the general label-skew problem. If the authors want to keep this masking trick in their paper, I believe its restrictions must be stated much more clearly. Should the authors do that, I will raise to 7.

---

> > > > > > ### Author Response · Authors · 2020-11-24
> > > > > > **Response**
> > > > > >
> > > > > > Thank you for the constructive comments. We address each of the concerns below.
> > > > > >
> > > > > > 1.  We delete sentence "Thus, this method greatly reduces the risk of leaking private data" in the latest revision. We agree that there are many other attacks can be performed other than reconstructing original dataset. Thanks for pointing that out. We mention in the latest revision that "There exist privacy concerns about calculating global statistics cumulatively and we hope to address those issues in the future work".
> > > > > >
> > > > > > 2.  We assume the change point of sub-task is a prior knowledge of the user. This assumption is common in other learning framework like continual learning (task-incremental learning). There also exist techniques to detect the change point of task (class-incremental learning) but this topic is still very challenging.
> > > > > >
> > > > > > 3.  We address the point about general label-skew and other non-IID partition by mentioning 'balanced' non-IID data partition and feature skew (vertically split) in our last revision. In the latest revision, we change the sentence 'To address statistical heterogeneity, we propose a “Masking Trick” and demonstrate that personalization is unnecessary for non-IID data in classification problems' to 'To address statistical heterogeneity, we propose a “Masking Trick” for balanced non-IID data partition in classification problems.' We agree that balanced non-IID label skew is narrow. We merely follow the experiments of FedAvg, LG-FedAVg to demonstrate a simple trick for balanced non-IID data partition problem. Masking trick is not enough to cure unbalanced non-IID data partition problem alone but can be used as a pre-screening of non-IID data partition. If the prior knowledge of which classes of data would appear is known, masking trick can be used to zero out needless outputs first before we apply other methods to tackle unbalanced non-IID data partition.

---

> > > > > > > ### Comment · AnonReviewer4 · 2020-11-24
> > > > > > > **Good work**
> > > > > > >
> > > > > > > Thx for addressing these things. I feel like we are on the same page.
> > > > > > > I will update my score to 7.

---

### Official Review · AnonReviewer1 · 2020-10-26
**A new FL algorithm and few tricks to improve heterogeneous FL.**

**Rating:** 7
**Confidence:** 5

**Review:**

This work presents a novel FL algorithm named HeteroFL (the name might sounds weird to some peoples) and 3 different simple methods to improve FL in heterogeneous conditions (i.e. both in term of clients and data partitioning). These tricks are: 1. A revised batchnormalisation; 2. a pre-activity scaling; 3. a masked loss (i.e only consider local classes)  to help with non-IID datasets. All these modifications have been tested on 3 different datasets and 2 different tasks. From the results, we can see that the proposed approach works better. Although, it is not clear from where the benefit comes.

My biggest concern with this paper lies in the complexity of the problem raised vs the lack of analysis of the proposed solutions. Appart from HeteroFL that is definitely a very nice idea, the small tricks try to address very important concerns related to FL. 1. What do we do with running statistics?; 2. How do we manage highly non-iid partitioning?. Unfortunately, the given solutions appear to be small "tricks" or "fixes" without any real theoretical or empirical insights. In my opinion, each problem should be detailed and discussed in a standalone paper. In this extent, reading this paper is a little bit like: "We present this method (HeteroFL), that works pretty well if we add these little fixes to very important problems". But we have no-idea on how these little fixes actually help HeteroFL (and thus could also help FedAVG, FedPROX etc etc).

However, it is worth noting that the core idea of this paper: HeteroFL, is a very simple and elegant way to deploy FL on an heterogeneous client set.

Pros:
+ HeteroFL is a very nice idea to deploy FL with an heterogeneous set of clients, and it seems to work well.
+ While the reasons aren't clear, the 3 proposed methods to stabilise and enhance the training process could help with bigger FL questions.
+ The paper is self-contained.

Cons:
- some claims are wrong: "the clients with the lowest capabilities will not be the bottleneck of FL’s performance" -> Well, according to the results, adding smaller clients (i.e. less powerful) always harms the performance. "The results show that HeteroFL can boost clients’ performance with low computation and communication capabilities by allowing the training of heterogeneous models with larger computation complexities." -> This sentence is unclear and leads to a wrong statement. HeteroFL isn't boosting the performance of small clients. Indeed, we can expect that small clients aren't even able to train the model, so HeteroFL is allowing us to train on such clients, but it is not boosting the performance. According to the results, smaller models = worst performance.
- sBN, scaling and masked loss "seem" to help, but it absolutely unclear to which extent. Also, why are the statistics finally uploaded to the server for inference (isn't this a leak of information ?). The theoretical motivation of the Scaler with the dropout analogy isn't clear at all. I'm pretty sure that it is mostly a matter of re-phrasing.

Remarks:
- Eq. 1, 2 and 3 are a bit hard to read. Is the "\" symbol used to denote integer divisions ? From a first read, it is not clear how weights are aggregated in intermediate complexity levels.
- The explanation of the effect of the proportionality is really unclear. Therefore, it is very hard to understand what Figure 2 (and the others) are about ... "To demonstrate the effect of proportionality of clients with various computation complexity levels, we interpolate from 10% to 100% with step size 10% of global model proportionality. For example, a − b means to interpolate between a and b models starting from 10% of clients assigned level a and 90% of clients assigned level b to 100% level a clients. " -> This is very hard to understand. What is global model proportionality ?
- Most of the Figures aren't black and white compatible (all the curves are almost impossible to distinguish).
- Section 3 should be splitted in 4 parts. One for each proposed method.

---

> ### Author Response · Authors · 2020-11-17
> **Response**
>
> Thank you for the positive comments. We address each of the six concerns below.
> 1. There may be some misunderstandings on how we boost clients' predictive performance. We are boosting weak clients instead of small models. In the FedAvg, if some weak clients can only train a small model, say our model 'e', due to computation and communication constraints, all the other clients have to train the same small model 'e' in order to average the model parameters from weak clients. The weak clients are thus the bottleneck of the overall performance of Federate learning. What we advocate is that each local client can train models corresponding to their personalized computation and communication capabilities. Then, the aggregated global model is deployed for each local client for future inference.
> The takeaway of Table 2 is that a weak learner that can only train a small model 'e' (on CIFAR10)(77\%) can boost its performance to 'c-e' (86.88\%), 'b-e' (89.10\%), or 'a-e' (90.29\%), which are close to the scenario where all the learners are strong, namely c(87.55\%),  b(89.82\%), or a(91.99\%).
> In particular, in 'c-e', 'b-e', or 'a-e', half of the clients are trained with larger models 'c', 'b', or 'a', while the other half are trained with the model 'e'. During the testing stage, only the aggregated global models 'c', 'b', or 'a' are used. Although weak clients train smaller models 'e', they will test with the largest models 'c', 'b', or 'a' to gain a better performance.
> 3. To address the concern of the usefulness of sBN, scaling, and masked loss, we provide a comprehensive ablation study in the revision.
> Our latest results show that sBN outperforms InstanceNorm, LayerNorm, and GroupNorm.
> During the training, sBN simply does a batch-wise normalization without tracking running estimates to ensure that the features before meeting the affine parameters will have a zero mean and unit variance. Thus, the local training and global aggregation of model parameters are stable even for models of different sizes. And from extensive experiments, we found that if the batch size for testing is small (e.g., only one), the test performance will degrade. Transmitting the statistics will help improve predictive performance.
> As to privacy, we agree that there will be some information leakage from transmitting the statistics.
> Nevertheless, we argue that the one-time transmission of feature statistics is typically not enough to reconstruct the original data. Otherwise, it is risky to share trained models with others, and the use of federated learning will be questionable.
> On the other hand,  the transmission of running statistics as used in the vanilla BN will likely to cause privacy issues. That is why we advocate the one-time transmission of feature statistics of a finalized model.
> In the inverted dropout, randomly sampled subnetworks are trained, and the features are multiplied according to the dropout rate.
> In our context, the heterogeneous models are reminiscent of the subnetworks in the dropout.
> The inverted dropout trains models with small widths but tests the complete model. We use the scaler module to operate in a similar manner as in the inverted dropout during the training; we test the global network in the inference stage. Our latest ablation study shows that the scaler module is indeed beneficial.
> 4. The slash symbol denotes the set difference. In our notation, the $W_i^p$ is a matrix/tensor, $W^t_g[:d_m,:k_m]$ is the upper left submatrix of $W^t_g$, with a size of $d_m \times k_m$, and the $W^{p-1,t+1}_g \setminus W^{p,t+1}_g$ denotes the set of elements (parameters) in $W^{p-1,t+1}_g$ but not in $W^{p,t+1}_g$. Each parameter will be averaged from those clients whose allocated parameter matrix contains that parameter. Thus, a model of an intermediate complexity will have parameters fully averaged with all the other larger models but partially with smaller models (according to the corresponding upper left submatrix).
> 5. The original wording of 'global model proportionality' was confusing, and we have removed it. The model proportionality indicates the relative number of strong and weak clients in our experiments.
> We interpolate from 10\% strong to 100\% strong clients with a step size of 10\%. For example, suppose that we have 100 clients. There are 10 of them trained with the model 'a', and 90 of them trained with the model 'e'.
> The averaged number of model parameters in this case is 0.1*(size of model 'a') + 0.9*(size of model 'e'). In Tables 1-3, the splitting ratio is 50\%, and the complexity of each particular client can dynamically vary at each communication round. We have elaborated more on this part in the revision.
> 6. We will update the line types and markers in the next revision (after incorporating new comments).
> 7. According to your suggestion, we have now split Section 3 into four parts in the revised paper.

---

### Official Review · AnonReviewer2 · 2020-10-28
**Promising results, but the paper and the work should be improved.**

**Rating:** 6
**Confidence:** 5

**Review:**

This paper proposes a framework named HeteroFL to address heterogeneous clients equipped with very different computation and communication capabilities.

The proposed methods were examined on three datasets for image classification and language modelling tasks.

Experimental results show that the proposed method boosts accuracy of the baseline FedAvg.

However, there are various undefined notation and missing details in the paper. Therefore, the paper is not easily readable and requires readers to make assumptions to fill the gaps. Some of the proposed strong claims should be supported and verified either theoretically or experimentally. In addition, experimental analyses should be improved by comparing the proposed method with the other related work.

More detailed comments and suggestions are as follows:

- Please use different notation for matrices and sets. For instance, $W$ denotes both a matrix/tensor and a set in the paper, which causes a problem in equations (1)-(3).

- In equation (1), does W_i^p denote a set or a matrix/tensor?

- What do W^t_g[: d_m, : k_m] and W^{p−1,t+1}_ g \ W^{p,t+1}_g denote?

- Could you please elaborate why "small local models can benefit more from global aggregation by performing less global aggregation for part of larger local model parameters."? That is, what are the benefits, and how do you assure that this approach enables to have these benefits?

- Please explain how you calculate statistics of hidden representations from local data after the model converges more precisely. How do you determine the convergence criteria?

- It is stated that "Local models only upload their statistics for once after optimization is completed." What are these statistics and how are they used when they are uploaded to the server?

- r^{p-1} is a scalar constant. Therefore, Scaler module scales feature activations by a constant. How does this help training?

- Please define \phi() (activation layer) more clearly. Do you refer to a non-linear activation function?

- Please define local capabilities information and L_{1:k} and L_m.

- Please describe details of architectures of the CNN, PreResNet18 and Transformer used in the experiments.

- Please explain what "Standalone" denotes in the tables.

- Please explain how you construct complexity levels in detail.

- Please explain the masking operation used in Masked Cross-Entropy Loss. More precisely, how do you mask out the output?

- The proposed method follows similar motivation and approaches of split learning and federated-split learning methods.
Could you please elaborate similarity/difference between your proposed methods and these methods?
A comparative experimental analysis would be also helpful to explain the novelty over these methods.

Following the rebuttal:

I checked comments of other reviewers and response of authors. Most of my questions were addressed in these responses. Therefore, I improve my rating.

---

> ### Author Response · Authors · 2020-11-17
> **Response**
>
> Thank you for the constructive comments. We address each of the concerns below.
>
> 1.  (for 1-3 bullet points) We denote the $W_i^p$ as a matrix/tensor. The $W^t_g[:d_m,:k_m]$ denotes the upper left submatrix with a size of $d_m \times k_m$.  Also, $W^{p-1,t+1}_g \setminus W^{p,t+1}_g$ denotes the set of elements included in $W^{p-1,t+1}_g$ but excluded in $ W^{p,t+1}_g$. We have clarified the notation in the revision.
> 4.The benefits are for weak clients with small local models. In the classical FedAvg, if some weak clients can only train a small model 'e' due to computation and communication constraints, all the other clients have to train the same small model 'e' to average the model parameters from weak clients. The weak clients are thus the bottleneck of the overall performance of Federate learning. We propose that each local client train models corresponding to their personalized computation and communication capabilities. Then, the aggregated global model is deployed for each local client for future inference.
> The takeaway of our experiment results in Table 2 is that a weak learner that can only train a small model 'e' (on CIFAR10)(77\%) can boost its performance to 'c-e' (86.88\%), 'b-e' (89.10\%), or 'a-e' (90.29\%), which are close to the scenario where all the learners are strong, namely c(87.55\%),  b(89.82\%), or a(91.99\%).
> In particular, in 'c-e', 'b-e', or 'a-e', half of the clients are trained with larger models 'c', 'b', or 'a', while the other half are trained with the model 'e'. Only the aggregated global models 'c', 'b', or 'a' are used during the testing stage. Although weak clients train smaller models 'e', they will test with the largest models 'c', 'b', or 'a' to gain better performance.
> 5. The local statistics of the sBN layer is computed for the local dataset first and then uploaded to the server. The server can aggregate those statistics based on the number of data samples in each local client. We specify a maximum number of epochs to converge, which is a standard implementation way, e.g., in the original FedAvg and LG-FedAvg papers.
> To show the convergence, we also included the learning curves (as evaluated by out-sample performance) in Figures 9-11 of the appendix.
> 6. The statistics uploaded to the server include the means and standard deviations of the features entering the sBN layers in our networks. Once they are uploaded, the server aggregates them and then broadcasts them to local clients. During the training, clients do not transmit any statistics. The server only gathers them after the training process is finished.
> We need to transmit those statistics because the batch size for testing may be different from the batch size for training. From extensive experiments, we found that if the batch size for testing is small (e.g., only one), the test performance will degrade. Transmitting the above statistics will help improve the testing performance.
> 7. The scaler module helps the training because the feature norms from models of different sizes can vary significantly. When we want to combine networks with different widths, it helps to train under a balanced scale of features. To further illustrate this point, we have provided a comprehensive ablation study in the revision (included in Tables 4 and 5 of the revised paper).
> 8. $\phi()$ (activation layer) is indeed the activation function commonly used, e.g., the ReLU().
> 9. The $L_m$ is an abstraction of the computation and communication capabilities of a local client $m$. Once this information is communicated to the server, the server can know the model complexity that should be allocated to the client. We have clarified this in the revision. Those arguments were made to address a realistic user scenario. The $L_{1:K}$ denotes the set of $L_1,\ldots,L_K$.
> 10. The hyperparameters of the CNN, PreResNet18, and Transformer architectures are included in Table 6 of the appendix. We will release our source codes once the paper is published.
> 11. The 'Standalone' in the tables means there is no communication between clients and the server. We have elaborated more on our experimental settings in the revision.
> 12. The complexity levels are associated with the shrinkage ratio. The parameter shape $d$ and $k$ is multiplied by $r^{p-1}$ where $p \in \{1,2,3,...\}$.
> We have tried other shrinkage ratios, and we found that it is most illustrative to use the discrete complexity levels 0.5, 0.25, 0.125, and 0.0625 (relative to the most complex model).  For example, model 'a' has all the model parameters, while models 'b' to 'e' have the effective shrinkage ratios 0.5, 0.25, 0.125, and 0.0625. We note that the complexity of 'e' is close to a logistic regression model. In practice, using a dictionary of discrete complexity levels provides convenience for coordination.

---

> > ### Author Response · Authors · 2020-11-17
> > **Response continue**
> >
> > 11. The masking is applied at the output of the classification network. For example, suppose that $y=NN(x)$, where $x$ is the data, and $y$ is the output. The cross-entropy loss is $L=CE(y, c)$, where $c$ is the label. If $y = [0.1, 0.2, 0.3, 0.4]$ for a four-class classification problem, but the local model only has data from the first two classes, then the masked output is $\hat{y} = [0.1, 0.2, 0, 0]$. The main idea is that the local model should not have a loss from the classes it does not observe. This technical trick was used to enhance the performance of non-iid FL. We have provided a comprehensive ablation study in the recent revision (in the current Table 5).
> > 12. Our method is different from SplitFed because we do not rely on the server to train part of the network.
> > Another difference is that we do not need to transmit gradients. The gradient information communicated in SplitFed may reveal sensitive data information according to a series of recent work (e.g., the 'deep leakage from gradients'). Our modification of the training procedure from FedAvg is minimal, which can be readily adapted to existing applications.

---

### Official Review · AnonReviewer3 · 2020-10-31
**Well motivated idea, well written**

**Rating:** 6
**Confidence:** 3

**Review:**

This paper proposes a new federated learning framework called HeteroFL, which supports the training of different sizes of local models in heterogeneous clients. Clients with higher computation capability can train larger models while clients with less computation capability train smaller models, and all these model architectures belong to the same model class. This approach dramatically benefits clients with limited computation capability and fully exploits their computation power.

Strengths:
1. The paper is well-motivated. The communication and computation problem does exist in federated learning, which made the proposed approach practical to apply.
2. The idea is novel enough. As far as I know, no other papers exploit the potential of model heterogeneity in federated learning.
3. The experiment is solid and rigorous enough to support the main idea. The authors use three different models and three corresponding datasets to conduct the experiments. The results show that model heterogeneity is promising, which outperforms state-of-the-art federated learning approaches.
4. The paper is well written. The authors organize the whole paper concisely and comprehensively, which makes the paper easy to read.

Weakness:
1. Although static batch normalization and Masked Cross-Entropy Loss are not the main contributions of this paper, I think it's more persuasive to prove their effectiveness by comparing it with baseline experiments instead of simply applying it.
2. The experiment setting should be elaborated more, such as the hyperparameters.

Also, I have some questions regarding the paper content:
1. Why the parameter size of Standalone, FedAvg, and LG-FedAvg is smaller than many other models in Table 2? Intuitively, Standalone, FedAvg, and LG-FedAvg use complete model architecture, so the parameter size should be the largest. However, it is less than models like a and b in Table 2.
2. How did you choose the shrinkage ratio? Have you compared the results of different shrinkage ratio? And also, Have you tried other methods of shrinking models?

---

> ### Author Response · Authors · 2020-11-17
> **Response**
>
> Thank you for the positive comments. We address each of the three concerns below.
>
> 1.  According to your suggestions, we have provided a comprehensive ablation study in the revision (currently Tables 4 and 5 in the revised paper). Also, we have elaborated more on the experimental settings in the revision. In particular, the hyperparameters are summarized in Table 6 in the appendix.
> 2.  It is because the tabulated results of Standalone, FedAvg, and LG-FedAvg were state-of-the-art results gathered from the LG-FedAvg paper. But in our experimental studies, we considered more complex models compared with the existing work.
> In particular, the baseline models in LG-FedAvg used MLP (on MNIST) and CNN (on CIFAR10). In terms of the number of parameters, our models 'a-e' (on MNIST) and 'b-e' (on CIFAR10) are comparable with those baselines. In terms of the FLOPs, our model 'd-e' (on MNIST and CIFAR10) can be compared with those baselines. The single-letter models 'a', 'b', 'c', 'd', 'e' are our implementations of the FedAvg equipped with the sBN and Masking CrossEntropy.
> Our ablation study shows that our the sBN and Masking CrossEntropy can significantly improve the results for all the above model configurations.
> The takeaway of Table 2 is that a weak learner that can only train a small model 'e' (on CIFAR10)(77\%) can boost its performance to 'c-e' (86.88\%), 'b-e' (89.10\%), or 'a-e' (90.29\%), which are close to the scenario where all the learners are strong, namely c(87.55\%),  b(89.82\%), or a(91.99\%).
> In particular, in 'c-e', 'b-e', or 'a-e', half of the clients are trained with larger models 'c', 'b', or 'a', while the other half are trained with the model 'e'. Only the aggregated global models 'c', 'b', or 'a' are used during the testing stage. Although weak clients train smaller models 'e', they will test with the largest models 'c', 'b', or 'a' to gain better performance.
> 3.  We have tried various shrinkage ratios, and we found that it is most illustrative to use the discrete complexity levels 0.5, 0.25, 0.125, and 0.0625 (relative to the most complex model).  For example, model 'a' has all the model parameters, while models 'b' to 'e' have the effective shrinkage ratios 0.5, 0.25, 0.125, and 0.0625. We note that the complexity of 'e' is close to a logistic regression model. Our experiments indicated that the ratio can be arbitrary between (0, 1] and dynamically change. In practice, using a dictionary of discrete complexity levels are convenient for coordination purposes.
> Yes, we have also tried many other methods of shrinking models. According to our literature search, the common ways of shrinking a model are to reduce its depth, width, and shape, as shown in the paper of EfficientNet. We found that reducing the depth and shape is less suitable for aggregating a global model during the testing stage than reducing width.
> Dropout can be seen as a way to reduce the width, and the inverted dropout originally inspired our scalar module.

---

### Author Response · Authors · 2020-11-17
**ablation study and clarification**

Thank you all for the valuable reviews. The main concern raised in the review comments is that we lack demonstrations of the effectiveness of scaler, sBN, and Masking CrossEntropy. We tried many choices such as vanilla BN, InstanceNorm, no scaler, and no masking during our earlier development. We have worked on a comprehensive ablation study in the past week to address the reviewers' concerns. They are included in Tables 4 and 5 (in supplementary) of the current revision. Our current ablation results show that our proposed methods are indeed beneficial.

We also want to clarify that though weak clients are trained with smaller models, and strong clients are trained with larger models, \textbf{there is only one single global model, which is the largest aggregated model to test}. The weak clients do not test with the smaller subnetworks after aggregation. It is the primary reason why 50\% weak clients can significantly boost their performance. We mainly address the challenge of computation and communication bottleneck in the training stage, and we assume that each local client can test with the largest model.

---

### Public Comment · ~Jean_Du_Terrail1 · 2020-11-17
**Notes for the PC**

I wanted to point out that the idea of keeping batch-normalization statistics siloed during training, that the authors coined “static Batch-Normalization”, is definitely in the air. In fact, it has already been explored by the prior published work [1] (missing citation), although in this submitted paper statistics are communicated  at the end of the training, whereas [1] never communicates statistics. There is also a concurrent ICLR submission [2], where the authors not only keep the statistics local but also the trainable BN weights.  This diminishes the novelty claim but doesn’t invalidate the premise of the paper.

I also note, in contrast with AnonReviewer4, that Batch-Normalization is only replaced by GroupNormalizations and other alternatives in theory. In practice nearly all state-of-the-art networks still use Batch-Normalization, including the recent EfficientNets[3, 4, 5] and all ResNet variants such as ResNeSts [4].

[1] Siloed Federated Learning for Multi-Centric Histopathology Datasets, Andreux et al., Proceedings of DART/DCL@MICCAI 2020
[2] FedBN: Federated Learning on Non-IID Features via Local Batch Normalization, ICLR2021 submission
[3] EfficientNet: Rethinking Model Scaling for Convolutional Neural Networks Tan and Le, Proceedings of ICML 2019
[4] ResNeSt: Split-Attention Networks, Zhang et al., arXiv
[5] Adversarial Examples Improve Image Recognition, Xie et al., Proceedings of CVPR 2020
[6] Fixing the train-test resolution discrepancy: FixEfficientNet, Touvron et al., arXiv

---

> ### Author Response · Authors · 2020-11-17
> **Response**
>
> Thank you all for the valuable comments.
>
> 1. Compared to Siloed Federated Learning and FedBN, our method does not track running statistics  or affine parameters locally during training. We found that uploading vanilla BN running statistics for networks of different sizes will degrade the performance and cause privacy issues. Our primary consideration is to train local models of different sizes. We do not track local running statistics and only normalize the feature. Therefore, each communication round is independent and local learners may not keep their statistics. In particular, the sizes of local models may vary dynamically at each communication round, and the locally tracked statistics at the previous communication round may not be useful at the next communication round.  Therefore, we advocate training without tracking running statistics and only compute and upload them once the overall training process is finished. The affine parameters are uploaded and aggregated similarly to the weight matrix in HeteroFL at each communication round. Thank you for pointing out this related work, and we will cite this paper.
>
> 2. Indeed, in practice, we do not usually use other kinds of normalization methods. Our ablation study shows that sBN also outperforms well-known variations of normalization methods in various FL settings.

---

> > ### Comment · AnonReviewer4 · 2020-11-23
> > **BN has alternatives in FL**
> >
> > Hi Jean,
> >
> > my comment was not meant to imply that BN has viably alternatives. In practice it doesn't, you are right. ResNeSt and EfficientNets, etc. all use BN.
> > My comment was directed towards BN in the federated learning setting in particular, where it has been difficult to apply because there is no way to compute the BN statistics at the end of training across the whole data-set.
> > It is in the federated setting, where alternatives, such as Group Norm have been applied.

---

### Comment · ~Durmus_Alp_Emre_Acar1 · 2021-04-21
**A Naive Baseline in Heterogeneous Federated Learning**

Thank you for the very nice and clear work. I enjoyed reading it and had a few questions.

*Comparison against a naive baseline*: I performed decoupled FedAvg, with 50 architecture ‘a’ and 50 architecture ‘e’ models corresponding to complex and simple models. In other words, 50 architecture ‘a’ models were trained with FedAvg without leveraging 50 architecture ‘e’  models. Similarly, I also trained 50 architecture ‘e’ models using FedAvg without leveraging the 50 architecture ‘a’ models. I list the results out of the experiments in this [link](https://github.com/alpemreacar/ExperimentationHeteroFL/blob/main/Experimentation_HeteroFL.pdf).  As for the ResNet experiments, I used the provided code and utilized the proposed sBN layers. I believe my tabulated results closely match that in the paper. From what I see training in isolation is generally better than HeteroFL. The performance of the Naive Baseline appears to be consistently outperforming HeteroFL across different architectures.

I would appreciate your comments, and point out if I am going wrong in my understanding.

---

> ### Author Response · Authors · 2021-04-21
> **About naive baseline of using model 'a' and 'e' alone**
>
> Thanks for your intesert in our work.
> The main contribution of our work is that we provide a framework for learning heterogeneous local models based on computation capabilities. I do not quite understand the HeteroFL result shown in your table because I suppose you should compare decoupled 'a' and 'e' against HeteroFL 'a-e'.  In our paper, CIFAR10 (IID ,'a-e') produces 90.29 which is better than the naive baselines.
>
> We want to clarify that It is indeed possible that by ignoring 50% of data, only training model 'a' can perform as well as 'a-e'. However, this comparison is data and task dependent. Our goal is to show the possiblity of training local heterogeneous model rather than always outperforming the case where we just ignore some part of the data (maybe 20%? 30%? and Non-IID case?). This is the reason we show the interpolation results with various combinations of computation complexity levels.

---

> > ### Comment · ~Durmus_Alp_Emre_Acar1 · 2021-04-28
> > **More Experimental Results of the Naive Baseline**
> >
> > Thanks for the follow up.
> >
> > Based on authors comments, we performed extensive experiments to benchmark naive strategy of decoupled federated learning with HeteroFL. We tabulate results by varying different networks (ResNet, 3 conv. layers), different assignments (fixed vs. dynamic), and different numbers of complex (Architecture e) and simple devices (Architecture a). The results appear at this [link](https://github.com/alpemreacar/ExperimentationHeteroFL/blob/main/Experimentation_more_results.pdf). Two observations that we can summarize from our findings:
> >
> > (I) Our ResNet HeteroFL performance is consistent with those reported in ICLR paper,
> >
> > (II) Our decoupled FedAvg consistently outperforms HeteroFL uniformly across all the different experimental settings.
> >
> > We agree that HeteroFL presents a framework for training heterogeneous devices in the federated architecture. Nevertheless, we are somewhat confused, and perhaps the authors can help us understand our question: Shouldn’t the benefits of a framework be manifested in experiments? If not, perhaps we are missing the point here, and will appreciate a response.

---

> > > ### Author Response · Authors · 2021-04-29
> > > **About naive baseline**
> > >
> > > Thank you for your additional feedback.
> > > 1. We assume all the data must be used in our experiments because of an underlying mental experiment. For example, we have five groups of clients, and each of them has different computation capabilities. For a 10-class classification problem, if each group only has two data classes, then decoupled FL will produce five different binary classifiers. We conducted a simple experiment to demonstrate a less extreme case. For a7-d3 (Fix, Non-IID, resnet18, CIFAR10) experiment [Local Acc(Global Acc)], decoupled 'a' produces [90.1 (52.4)] and decoupled 'd' produces [86.6(48.1)], while our proposed HeteroFL produces [92.7(55.7)]. This result validates our assumption.
> > > 2. We conduct experiments for 30% of CIFAR10 data for centralized model 'a', and we get 90.3. Your naive baselines for decoupled 'a' shows that one of our major contributions, the sBN, provides a very strong baseline for FL even if the number of data samples is small.
> > > 3. The most ideal FL case 'a' (IID, resnet18, CIFAR10) with the sBN produces 91.2. It is foreseeable that 'a5-e5' cannot increase from 89 to 91 simply by training 0.4% (38K/9.6M) parameters of 'a'. If we switch to 'a5-b5' by training more parameters, we can improve from 89 to 90.
> > >
> > > Our experiments do have some limitations on IID experiments. Our experimental setup follows the FedAvg paper. We hope our work can inspire more research on model heterogeneity in FL.

---

### Decision · Program_Chairs · 2021-01-07
**Final Decision**

**Decision:**

Accept (Poster)

**Comment:**

The reviewers had a number of concerns which seem to have been addressed by the authors in the discussion phase.  All the reviewers are in favor of accepting the paper. The paper provides an interesting/novel idea for federated learning with heterogenous clients/devices.